# OXPHOS deficiencies affect peroxisome proliferation by downregulating genes controlled by the SNF1 signaling pathway

Jean-Claude Farre*, Krypton Carolino, Lou Devanneaux, Suresh Subramani*

Section of Molecular Biology, Division of Biological Sciences, University of California, San Diego, La Jolla, United States

**Abstract** How environmental cues influence peroxisome proliferation, particularly through organelles, remains largely unknown. Yeast peroxisomes metabolize fatty acids (FA), and methylotrophic yeasts also metabolize methanol. NADH and acetyl-CoA, produced by these pathways enter mitochondria for ATP production and for anabolic reactions. During the metabolism of FA and/or methanol, the mitochondrial oxidative phosphorylation (OXPHOS) pathway accepts NADH for ATP production and maintains cellular redox balance. Remarkably, peroxisome proliferation in Pichia pastoris was abolished in NADH-shuttling- and OXPHOS mutants affecting complex I or III, or by the mitochondrial uncoupler, 2,4-dinitrophenol (DNP), indicating ATP depletion causes the phenotype. We show that mitochondrial OXPHOS deficiency inhibits expression of several peroxisomal proteins implicated in FA and methanol metabolism, as well as in peroxisome division and proliferation. These genes are regulated by the Snf1 complex (SNF1), a pathway generally activated by a high AMP/ATP ratio. In OXPHOS mutants, Snf1 is activated by phosphorylation, but Gal83, its interacting subunit, fails to translocate to the nucleus. Phenotypic defects in peroxisome proliferation observed in the OXPHOS mutants, and phenocopied by the Δgal83 mutant, were rescued by deletion of three transcriptional repressor genes (MIG1, MIG2, and NRG1) controlled by SNF1 signaling. Our results are interpreted in terms of a mechanism by which peroxisomal and mitochondrial proteins and/or metabolites influence redox and energy metabolism, while also influencing peroxisome biogenesis and proliferation, thereby exemplifying interorganellar communication and interplay involving peroxisomes, mitochondria, cytosol, and the nucleus. We discuss the physiological relevance of this work in the context of human OXPHOS deficiencies.

*For correspondence:
jfarre@ucsd.edu (JC-F);
ssubramani@ucsd.edu (SS)

Competing interest: The authors declare that no competing interests exist.

## Editor's evaluation

This paper elegantly describes how the assembly, division and proliferation of peroxisomes are controlled by proteins/metabolites residing in 4 different subcellular compartments. Mitochondrial energy production is assisted by peroxisomal proteins and metabolites, which shuttle via the cytosol to mitochondria, where ATP is produced. ATP availability is transmitted by the nuclear shuttling of cytosolic proteins to regulate genes controlling peroxisome assembly, division and proliferation.

## Introduction

Like other subcellular organelles, peroxisomes divide and segregate to endow daughter cells with peroxisomes. Cells stringently regulate organelle number, volume, size, and content in response to environmental signals. Regulation of these properties, together with organelle dynamics and homeostasis, allows cells to endure metabolic or environmental stress, cope with the needs of cell division or differentiation, remove excess or damaged organelles by turnover, correct imbalances in organelle

segregation during cell division, or repopulate organelles with different enzymes upon switching to a new environment. The division of some organelles, such as the nucleus or the Golgi apparatus, is coupled to the cell cycle (*Sütterlin et al., 2002*). However, for others, such as mitochondria and chloroplasts, division is uncoupled from cell division (*Osteryoung and Nunnari, 2003*). Peroxisomes division can be coupled or uncoupled from cell division (*Lingard et al., 2008*). The decoupling of peroxisome division during the cell cycle and cell division is likely due to the ability of cells to also produce peroxisomes de novo (*Hettema and Motley, 2009*).

In yeast and other eukaryotes, three major pathways control peroxisome number. First, in constitutively dividing cells, peroxisomes divide by fission of pre-existing peroxisomes, a process we refer to simply as peroxisome division (*Guo et al., 2003*; *Veenhuis et al., 2003*). This process regulates peroxisome number in a geometric manner. A second pathway is where peroxisomes are induced to create many new organelles exponentially, a process we call 'peroxisome proliferation'. Finally, peroxisome number is also controlled by pexophagy, the selective degradation of peroxisome by autophagic processes (*Farré and Subramani, 2016*).

In yeasts, peroxisome division occurs both during constitutive growth and cell division. Peroxisome size and number are sensitive to peroxisomal metabolic pathways and the metabolites available (*Chang et al., 1999*; *Poll-The et al., 1988*; *Smith et al., 2000*; *Yan et al., 2005*). Changes in peroxisome number and size can also be induced by peroxisome proliferation, which generally occurs when cells are shifted to nutrients whose metabolism requires peroxisomes and their enzymes (*Veenhuis et al., 2003*; *Lazarow, 2003*). Finally, peroxisomal contents can also vary depending on the growth media whose metabolism requires different peroxisomal enzymes (*Veenhuis et al., 2003*).

Peroxisome division uses a machinery also required for mitochondrial fission (*Schrader, 2006*). In yeast, this machinery is comprised of the proteins Fis1, Mdv, and Caf4, as well as the dynamin-related GTPase, Dnm1 (*Kuravi et al., 2006*; *Motley et al., 2008*). The Pex11 family of proteins activates peroxisome division specifically (*Huber et al., 2012*; *Koch and Brocard, 2012*; *Koch et al., 2010*). These proteins are conserved in evolution (*Schrader, 2006*; *Aung et al., 2010*) and are responsible for diseases in humans and plants (*Thoms and Gärtner, 2012*; *Wang et al., 2015*; *Weng et al., 2013*). In *Saccharomyces cerevisiae*, while Pex11 promotes the division of peroxisomes already present in the cell, Pex25 initiates remodeling at the peroxisomal membrane to allow proliferation and Pex27 counters this activity (*Huber et al., 2012*).

Our current understanding of the mechanisms by which environmental cues activate peroxisome proliferation is rather limited (*Yan et al., 2005*; *Aung et al., 2010*; *Gurvitz and Rottensteiner, 2006*; *Scheckhuber, 2020*). In *S. cerevisiae*, peroxisome proliferation coincides with the transcriptional regulation, by nonfermentable oleate, of SNF1 complex-mediated induction of several peroxisomal β-oxidation and peroxisome proliferation genes (*Karpichev and Small, 1998*; *Smith et al., 2002*; *Rottensteiner et al., 1996*; *Rottensteiner et al., 2003*). The Snf1 protein – the ortholog of the mammalian AMP-activated protein kinase (AMPK) – is a heterotrimer of the Snf1 catalytic subunit, the Snf4 activation subunit (*Jiang and Carlson, 1997*), and one of three β subunits (Sip1, Sip2, and Gal83) that localize the SNF1 complex to different cellular compartments (*Jiang and Carlson, 1997*; *Yang et al., 1994*). Gal83 directs the SNF1 complex to the nucleus in a glucose-regulated manner and facilitates the physical interaction between SNF1 and nuclear transcription factors (*Vincent et al., 2001*). In yeast cells grown in the absence of glucose, Snf1 is active via phosphorylation within its activation loop at Thr210 (T210) primarily by Sak1, but also by Tos3 and Elm1 (*Elbing et al., 2006*; *Hedbacker et al., 2004*; *Hong et al., 2003*). The activated SNF1 complex inactivates the transcriptional repressors, Mig1 and Mig2 (*Vallier and Carlson, 1994*), via their phosphorylation (*Serra-Cardona et al., 2014*; *Treitel et al., 1998*) and export from the nucleus, thereby enabling activation by the transcriptional activator, Adr1 (*Young et al., 2003*).

Two additional transcriptional activators, Oaf1 and Pip2, regulate genes in response to oleate induction conditions (*Gurvitz and Rottensteiner, 2006*; *Rottensteiner et al., 1996*). However, these transcription activators have not been described in *P. pastoris* (now reclassified as *Komagataella phaffii*) (*Prielhofer et al., 2015*).

In contrast, the addition of glucose to cells (glucose repression) results in a reduction in ADP levels that causes the Glc7-Reg1 protein phosphatase to dephosphorylate T210 and thereby inactivate Snf1 (*Chandrashekarappa et al., 2011*; *Ludin et al., 1998*; *Mayer et al., 2011*).

Notably, this entire description of peroxisome induction and proliferation involves transactions of proteins and metabolites residing in three compartments – the cytosol, nucleus, and peroxisomes, with no direct involvement of mitochondria. Yet, it is clear that peroxisomes contact other subcellular compartments, including mitochondria (*Dakik and Titorenko, 2016*; *Mattiazzi Ušaj et al., 2015*; *Shai et al., 2018*). Furthermore, mutations in genes affecting peroxisome biogenesis also impair mitochondrial function and morphology (*Baumgart et al., 2001*; *Salpietro et al., 2015*), and the reverse is also true (*Fransen et al., 2017*).

We probed more deeply into the regulation of peroxisome content and proliferation in *P. pastoris*. During growth in glucose, most *P. pastoris* cells possess a few, small peroxisomes, with a limited lumenal content, but when cells are grown in oleate, numerous, small peroxisomes proliferate and are distributed throughout the cells, whereas in methanol medium the peroxisomes are large, less numerous, and clustered (*Gould et al., 1992*).

We reveal here the mechanism and players involved in the important interorganellar interplay involving the cytosol, nucleus, peroxisomes, and mitochondria by which peroxisomal and mitochondrial metabolites influence redox and energy metabolism in the two compartments, while also influencing peroxisome biogenesis and proliferation.

## Results

### Peroxisome proteins/metabolites influence peroxisome size

We reinvestigated whether peroxisomal metabolites influence peroxisome size and number in *P. pastoris*. Peroxisomes were labeled with the functional, fluorescently tagged, peroxisomal membrane proteins (PMPs), Pex3-GFP or GFP-Pex36, in several mutant strains grown in different carbon sources (*Figure 1*). We deleted key genes encoding enzymes of the fatty acids (FA) β-oxidation and methanol metabolism pathways, as well as a peroxin required for the import of peroxisome matrix proteins. Supporting previous findings, wherein intermediates of peroxisome metabolism, or the absence of certain peroxisomal enzymes, regulate the maturation and fission of the organelle (*Nguyen et al., 2006*; *Espeel et al., 1997*), the lack of 3-ketoacyl-CoA thiolase (Pot1), an enzyme responsible for the last step of FA β-oxidation, affected peroxisome proliferation exclusively in oleate (*Figure 1—figure supplement 1*). Similarly, the lack of alcohol oxidase (Aox) 1 and 2, the first enzymes in the methanol utilization pathway (MUT), affected peroxisome proliferation only in cells grown in methanol medium (*Figure 1—figure supplement 1*). Similarly, the lack of Pex5, responsible for the import of PTS1-containing enzymes (including some involved in the β-oxidation and MUT pathways), impaired peroxisome proliferation in both media (*Figure 1—figure supplement 1*). These results show that one or more peroxisomal intermediate metabolite/s induces peroxisome proliferation, in addition to previous reports regarding their effects on division.

The molecular target of the signaling pathway that triggers peroxisome proliferation has not yet been elucidated. We hypothesized that if peroxisomal proteins/metabolites regulate proliferation, they must communicate with the cytosol and/or other organelles to induce the lipid and membrane transfer needed for peroxisome growth (*Mahalingam et al., 2021*; *Reinisch and Prinz, 2021*). Additionally, there must be activation of the fission machinery needed to increase peroxisome numbers (*Huber et al., 2012*).

In *S. cerevisiae*, the final intraperoxisomal products of β-oxidation are NADH and acetyl-CoA, for cells grown in FA (*Figure 1B*, *Wanders et al., 2020*). The transport of reducing equivalents out of peroxisomes and the maintenance of the intraperoxisomal redox balance in *S. cerevisiae* is mediated by the malate/oxaloacetate shuttle during growth in oleate, and by both the malate/oxaloacetate and the glycerol-3-phosphate/dihydroxyacetone phosphate (G3P/DHAP) shuttles, during growth in glucose (*Al-Saryi et al., 2017*). The malate/oxaloacetate shuttle is coordinated by three NAD⁺-dependent malate dehydrogenases: the mitochondrial Mdh1 that is part of the TCA cycle, the peroxisomal Mdh3 which regenerates $NAD^+$ for FA β-oxidation, and cytosolic Mdh2, probably involved in the glyoxylate cycle (*Gabay-Maskit et al., 2020*). The G3P/DHAP shuttle is coordinated by two G3P dehydrogenases: the mitochondrial Gpd2 and peroxisomal/cytosolic Gpd1 (*Kumar et al., 2016*; *Ansell et al., 1997*). *P. pastoris* possesses only two malate dehydrogenases (designated MdhA or B) and one G3P dehydrogenase (GpdA). We recently reported that MdhA and GpdA are predominantly mitochondrial enzymes in all media analyzed, whereas MdhB is cytosolic during growth in glucose

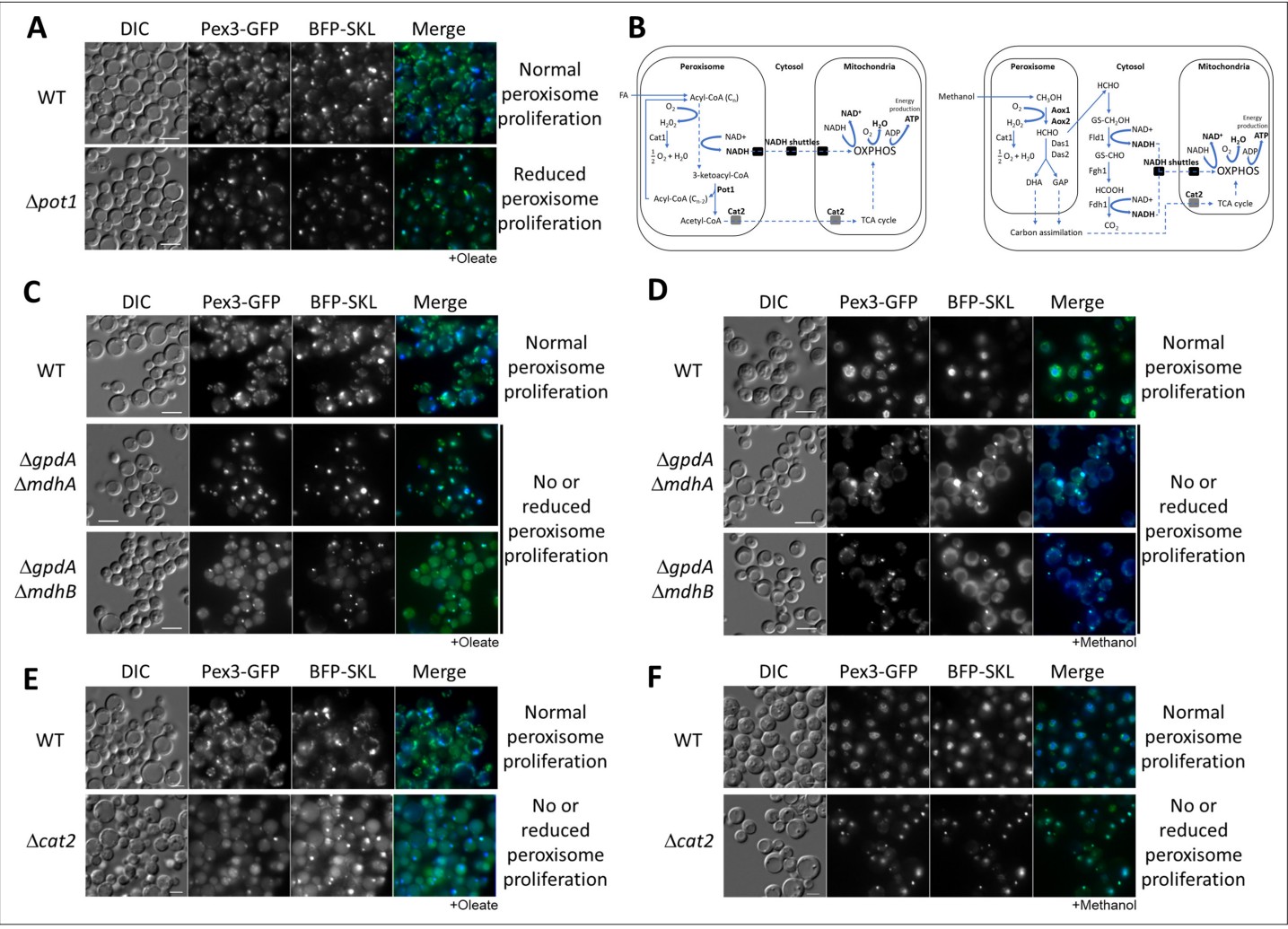

**Figure 1.** Peroxisome metabolites influence peroxisome size. (**A**) Fluorescence microscopy of WT and Δ*pot1* mutant cells expressing Pex3-GFP driven by the *PEX3* promoter and BFP-SKL driven by the *GAPDH* promoter, grown in oleate for 8 hr. (**B**) Brief description of FA β-oxidation, methanol metabolism, and NADH shuttling between peroxisomes and mitochondria. Cat1, catalase; Cat2, carnitine acetyl-CoA transferase; Pot1, 3-ketoacyl-CoA thiolase; Aox, alcohol oxidase; Das, dihydroxyacetone synthase; Fld1, formaldehyde dehydrogenase; Fgh1, *S*-formylglutathione hydrolase; Fdh1, formate dehydrogenase. (C–F) Fluorescence microscopy of WT, NADH shuttling and Δ*cat2* mutant cells expressing Pex3-GFP and BFP-SKL, grown in oleate and methanol for 8 hr, respectively. Bar: 5 μm.

The online version of this article includes the following figure supplement(s) for figure 1:

**Figure supplement 1.** Peroxisome metabolites influence peroxisome size.

**Figure supplement 2.** NADH-shuttling mutants influence peroxisome size.

or methanol, but is cytosolic/peroxisomal during growth in oleate (*Farré et al., 2021*). The transport of acetyl-CoA from the peroxisome to the mitochondria uses two pathways (*van Roermund et al., 1995*). The first involves peroxisomal conversion of acetyl-CoA into citrate by peroxisomal citrate synthase (Cit2), followed by citrate transport to mitochondria. The second pathway involves peroxisomal conversion of acetyl-CoA into acetylcarnitine by carnitine acetyl transferase (Cat2), which is then transported to mitochondria.

With the intent of blocking the shuttling of NADH and acetyl-CoA to the mitochondria, we made single and double deletions of components of both NADH shuttles, and the *CAT2* gene in *P. pastoris* and analyzed peroxisome status after 24 hr of induction in oleate medium using Pex3-GFP and BFP appended with a PTS1 (BFP-SKL) (*Figure 1C, E*). We observed no, or partial, peroxisome defects in the single NADH-shuttle mutants (*Figure 1—figure supplement 2*). Interestingly, in both double deletion strains, Δ*gpdA* Δ*mdhA* and Δ*gpdA* Δ*mdhB*, as well as Δ*cat2*, most of the cells grown in

oleate contained only a single, import-competent peroxisome (*Figure 1C*). The results for the NADH-shuttling mutants are consistent with their role in maintaining the intraperoxisomal redox balance and peroxisome proliferation during growth in oleate, but surprisingly the lack of Cat2, not directly implicated in redox balance, had a similar peroxisome proliferation defect.

As a methylotrophic yeast, *P. pastoris* uses methanol as a sole carbon source for carbon assimilation and energy production (*Figure 1B*). Peroxisomes are directly involved in methanol metabolism and similar to oleate, methanol induces peroxisome proliferation. Methanol is metabolized to formaldehyde, which diffuses to the cytosol and is oxidized by a dehydrogenase (Fld1) to formate, which is further oxidized by a second dehydrogenase (Fdh1) to carbon dioxide, yielding NADH. This NADH then shuttles to the mitochondria to maintain the cytosolic redox balance and to feed mitochondrial oxidative phosphorylation (OXPHOS). We analyzed the same yeast strains from *Figure 1C, E* in methanol and unexpectedly, we found the same peroxisome proliferation defects observed for oleate cultures (*Figure 1D, F*). The phenotypes of the NADH-shuttling mutants during growth in methanol are relevant because the redox reactions in this medium happen in the cytosol. Moreover, no major role of Cat2 during methanol metabolism was expected, but it might be feeding intermediates of methanol metabolism into the TCA cycle (*Figure 1B*; *Cámara et al., 2017*).

The similar phenotypes of the NADH-shuttling and *cat2* mutants in both media suggest that redox imbalance is not the main reason for peroxisome proliferation defects, but rather could be due to the absence of a molecule common to both pathways (NADH- and acetyl-CoA-shuttling pathways).

## Mitochondrial OXPHOS mutants also affect multiple peroxisome phenotypes

We hypothesized that NADH, which shuttles to the mitochondria during growth in methanol medium, or produced by the TCA cycle from peroxisomal metabolites, could trigger peroxisome proliferation. In *P. pastoris*, the NADH shuttled to the mitochondria is oxidized by the respiratory chain. The NADH dehydrogenase complex I (CI) is the main entry point for electrons into the respiratory chain. It directly oxidizes NADH to $NAD^+$ and concomitantly reduces coenzyme Q (CoQ, ubiquinone), which causes $H^+$ to be pumped out across the inner mitochondrial membrane, to contribute to the proton gradient that drives ATP synthesis (*Lasserre et al., 2015*).

To study the involvement of mitochondrial NADH oxidation in peroxisome proliferation and fission, we deleted two, nuclear-encoded genes encoding the mitochondrial complex I, the accessory subunit, Ndufa9, and the core subunit, NugM (*Lasserre et al., 2015*). Because none of these mutants have

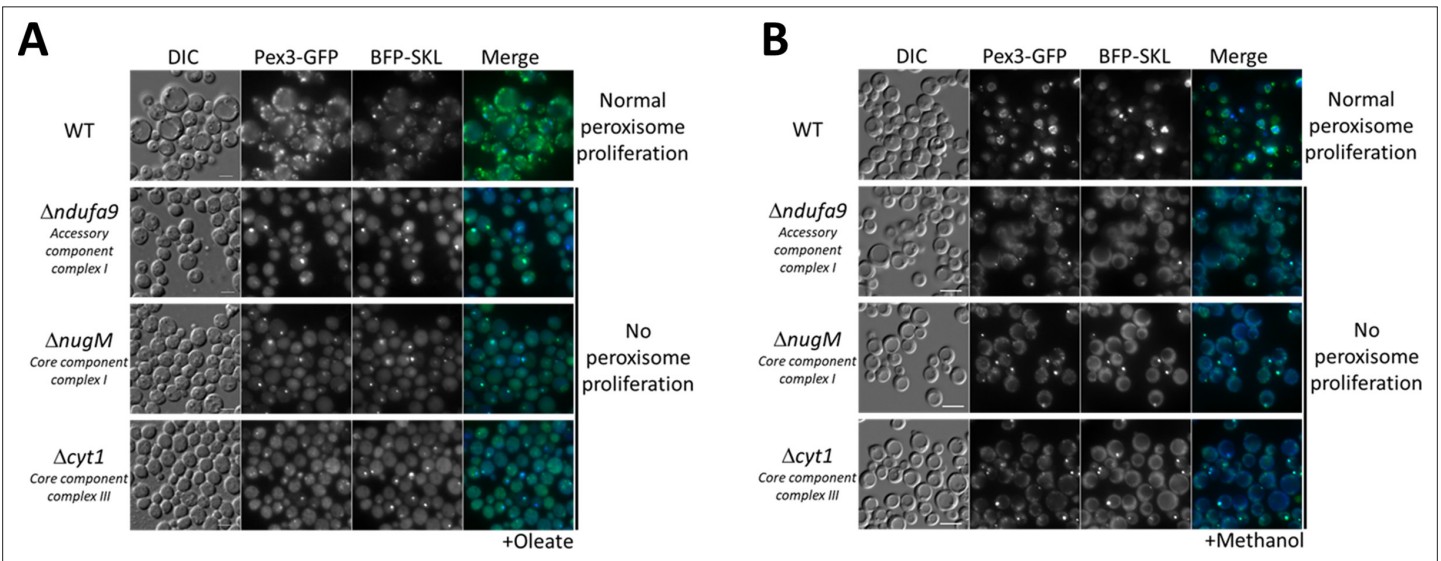

**Figure 2.** Dysfunctional mitochondria affect peroxisome proliferation. Fluorescence microscopy of WT, Δ*ndufa9*, Δ*nugM*, and Δ*cyt1* mutant cells expressing Pex3-GFP and BFP-SKL, grown for 16 hr in (**A**) oleate and 8 hr in (**B**) methanol medium. Bar: 5 μm.

The online version of this article includes the following figure supplement(s) for figure 2:

**Figure supplement 1.** Altered mitochondrial respiration in mitochondrial CI mutants.

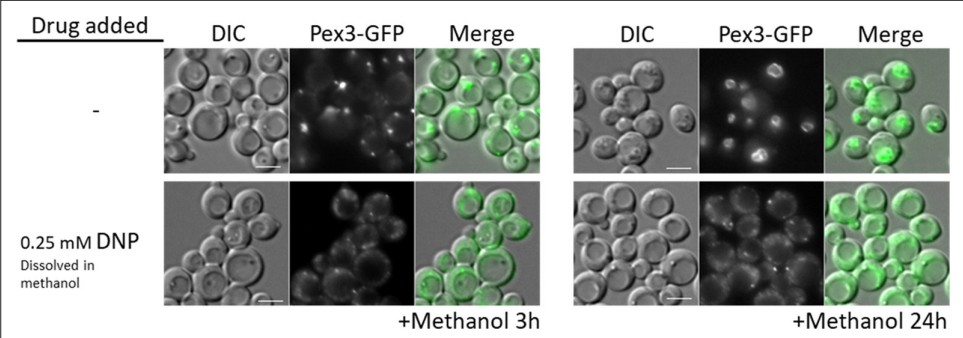

**Figure 3.** WT cells treated with the oxidative phosphorylation (OXPHOS) uncoupler, 2,4-dinitrophenol (DNP), share peroxisome proliferation defects with mitochondrial CI and CIII mutant cells. Fluorescence microscopy of WT cells expressing Pex3-GFP, grown in methanol medium, with or without 0.25 mM DNP. Bar: 5 µm.

The online version of this article includes the following figure supplement(s) for figure 3:

**Figure supplement 1.** Cells treated with the oxidative phosphorylation (OXPHOS) uncoupler, 2,4-dinitrophenol (DNP), share same peroxisomal protein expression defects with OXPHOS mutants, without affecting Snf1 phosphorylation.

been previously characterized in *P. pastoris*, we confirmed their necessity for the electron transport chain in response to metabolic substrates using Biolog's Yeast Mitochondrial Energy Substrate Assays (*Figure 2—figure supplement 1*).

Furthermore, as seen for NADH- and acetyl-CoA-shuttling mutants, we observed no increase in peroxisome number or size in either CI mutant in both media (*Figure 2*).

To differentiate a direct role of CI in peroxisome proliferation from the general role of electron transfer and proton translocation, we deleted a gene (*CYT1*, encoding a core subunit of CIII) encoding a downstream component of the OXPHOS complex, but not directly implicated in NADH oxidation. Like the CI mutants, the Δ*cyt1* mutant also blocked peroxisome proliferation in both media (*Figure 2*), indicating that peroxisome proliferation depends on a functional mitochondrial respiratory chain.

The similarity in peroxisome proliferation defects between mutants of CI and CIII suggested that the lack of ATP synthesis, and not the reduction of NADH, causes these phenotypes. To verify this hypothesis, we used the mitochondrial uncoupler, DNP, which separates the flow of electrons from the pumping of $H^+$ ions for ATP synthesis (*Pinchot, 1967*). We followed peroxisome proliferation in methanol after 3 and 24 hr, using WT cells expressing Pex3-GFP (*Figure 3*) and found that DNP fully abolished peroxisome proliferation, confirming that ATP synthesis is necessary for peroxisome proliferation.

## The OXPHOS effect on peroxisome proliferation acts via the Snf1 kinase pathway in *P. pastoris*

Due to the lack of peroxisome proliferation, we checked the relative protein abundance of the key peroxisome division factor, Pex11 (Pex11-2HA expressed from its own promoter, $P_{PEX11}$) after 4 hr of induction in oleate medium (*Figure 4A*). Remarkably, we did not detect Pex11-2HA in Δ*nugM* cells, suggesting that an inactive SNF1 pathway could explain the OXPHOS deficiency. In *S. cerevisiae*, the proliferative capacity of peroxisomes coincides with the FA-responsive transcriptional regulation of many genes encoding peroxisomal proteins, including Pex11 (*Hedbacker and Carlson, 2008*). Many such genes are repressed in glucose (*Kayikci and Nielsen, 2015*; *Kim et al., 2013*) and derepressed in oleate (*Saleem et al., 2008*) in a manner that depends on the SNF1 signaling pathway.

In *P. pastoris*, the regulation of peroxisomal genes involved in β-oxidation has not been studied much. However, many studies have focused on the regulation of the *AOX1* promoter ($P_{AOX1}$) during growth in glucose, glycerol, and methanol media (*Shi et al., 2018*; *Vogl et al., 2018*; *Wang et al., 2016a*; *Lin-Cereghino et al., 2006*). Similar to β-oxidation genes, *AOX1* is repressed during growth in glucose and strongly induced by methanol. The transcriptional activator, Mxr1 (which shares sequence and functional homology with *S. cerevisiae* Adr1) (*Lin-Cereghino et al., 2006*; *Hou et al., 2020*) and the transcriptional repressors, Mig1, Mig2, and Nrg1 (*Shi et al., 2018*; *Wang et al., 2016b*) (involved

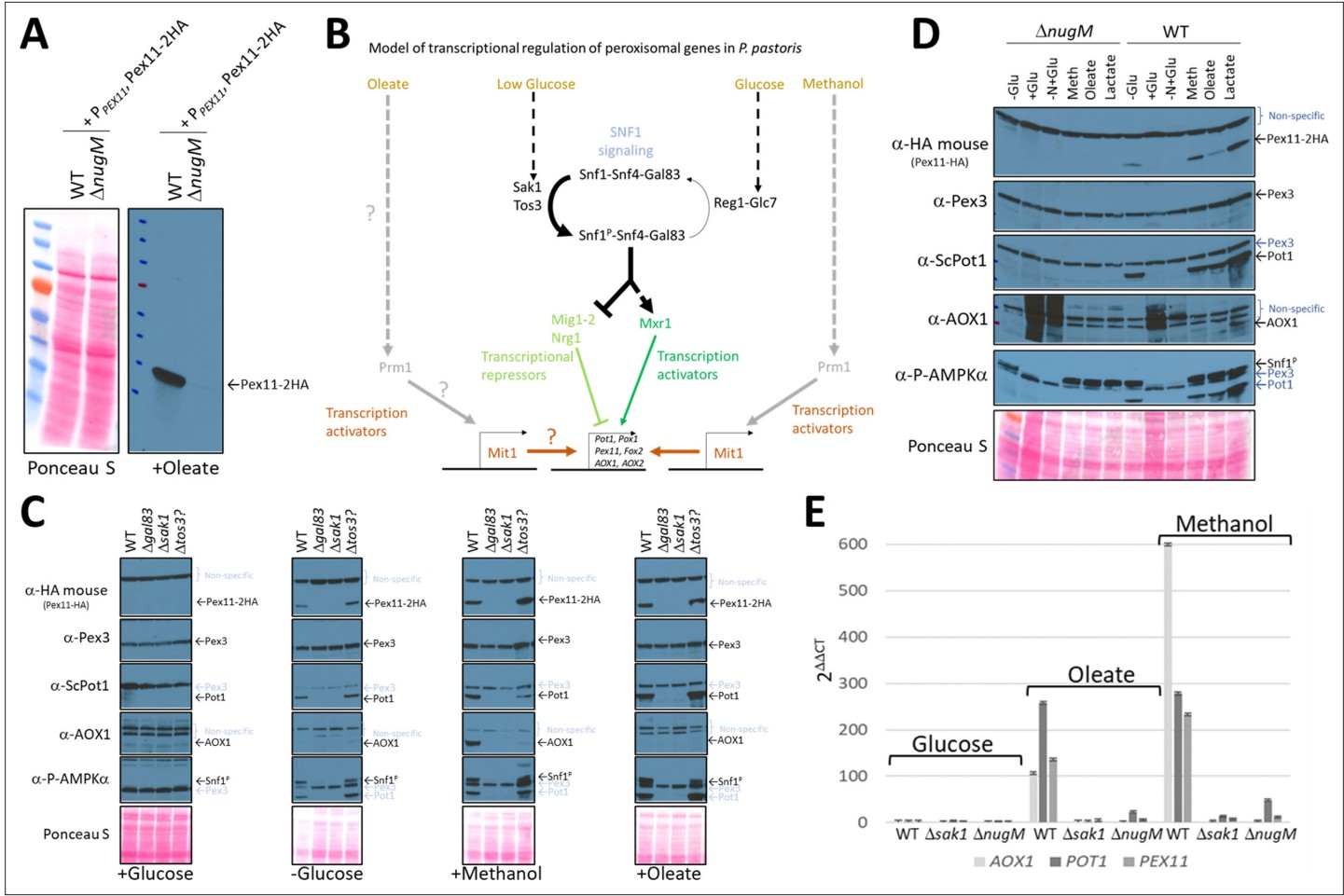

**Figure 4.** Oxidative phosphorylation (OXPHOS) mutants, like SNF1 mutants impair Pot1, Aox1, and Pex11 expression. (**A**) Western blot of Pex11-2HA visualized with anti-HA antibodies in WT and Δ*nugM* mutant cells. (**B**) Model of transcriptional regulation of peroxisome genes regulated by SNF1 signaling. ? denotes unknown pathway for induction of Mit1 in oleate. (**C, D**) Western blots of several peroxisomal proteins and the phosphorylated form of Snf1 in WT, SNF1 complex mutants and/or Δ*nugM* mutant cells. Nonspecific bands and signal arising from previous western blots are indicated in blue font. Ponceau S staining was used as a loading control. (**E**) Relative expression of *PEX11*, *POT1*, and *AOX1* after 4 hr incubation in the indicated carbon source using 18S ribosomal RNA (18S-rRNA) for qPCR normalization, and WT in glucose medium as reference and the $2^{-\Delta\Delta CT}$ method for the analysis (*Rao et al., 2013*).

The online version of this article includes the following figure supplement(s) for figure 4:

**Figure supplement 1.** Peroxisomal protein expression under different environmental conditions and validation of the phospho-AMPKα (Thr172) antibody in *P. pastoris*.

in glucose repression), regulate the derepression from glucose in *P. pastoris* (**Figure 4B**). As in *S. cerevisiae*, these transcription factors are most probably regulated by the SNF1 pathway, as suggested by a high-throughput screen implicating Sak1, the primary Snf1-activating kinase, and Gal83, the β-subunit of the SNF1 complex, in the expression of AOX (**Shen et al., 2016**).

In *P. pastoris*, the *S. cerevisiae* transcription activators, Pip2 and Oaf1, have not been found by in silico studies. However, methanol and glycerol activate at least two transcription factors, Mit1 and Prm1, which induce enzymes required for methanol utilization (MUT), but not for peroxisome proliferation, by a mechanism resembling Pip2-Oaf1 activation (**Wang et al., 2016a**; **Sahu et al., 2014**).

We analyzed Pex11-2HA, Aox1, Pot1, and Pex3 protein levels under different carbon induction conditions using available kinase deletions strains affecting the SNF1 signaling pathway (**Figure 4**, **Figure 4—figure supplement 1**; **Shen et al., 2016**). As expected, in WT cells, Pex11-2HA, Pot1, and Aox1 were not detected in glucose medium, but glucose derepression (-Glucose) induced Pex11-2HA and Pot1, but not Aox1, which has no role during the growth of WT *P. pastoris* in oleate. Glucose derepression, when combined with the addition of methanol, was needed to observe Aox1 expression in

WT cells in our cultivation conditions, indicating a more complex regulation, such as the activation by methanol of Mit1 and Prm1 (*Wang et al., 2016a*).

We examined Snf1 activation by phosphorylation next because glucose derepression acts via the SNF1 pathway. The amino acid sequence near the activation loop of *P. pastoris* Snf1 (including Thr 171) is identical with that in human AMPKα and *S. cerevisiae* Snf1 (*Yan et al., 2018*). Thus, a phospho-AMPKα (Thr 172) antibody designed to correspond to the residues surrounding Thr 172 of human AMPKα detects the active form of *P. pastoris* Snf1 (*Figure 4—figure supplement 1*). As seen in other organisms, *P. pastoris* Snf1 was activated in WT cells in response to glucose limitation by phosphorylation of Thr 171 of its catalytic subunit (*Figure 4C*). A mutant of a putative Tos3 kinase (Tos3?; UniProt gene name: PAS_chr1-3_0213), with weak homology to its *S. cerevisiae* counterpart, obtained from the *P. pastoris* kinase deletion collection (*Shen et al., 2016*), was not required for Snf1 activation, nor for the downstream SNF1 regulation, indicating that PAS_chr1-3_0213 is not required for SNF1 signaling or for peroxisomal protein expression. In contrast, Gal83 and Sak1 were essential for Snf1 phosphorylation and the expression of Pex11-2HA, Pot1 and Aox1 in every condition tested, confirming their major role in the SNF1 signaling pathway for the expression of some peroxisomal proteins.

We also analyzed Snf1 activation and expression of the same proteins in the Δ*nugM* strain using the same experimental conditions, and remarkably we observed similar expression defects as those observed for Δ*sak1* and Δ*gal83* strains, despite phosphorylation of Snf1 (*Figure 4D*). We previously described that DNP causes a similar peroxisome proliferation defect as the OXPHOS mutants (*Figure 3*), and like the Δ*nugM* mutant also affects the expression of Pex11, Pot1, and Aox1 during methanol cultivation, despite phosphorylation of Snf1 (*Figure 3—figure supplement 1*). Thus, DNP addition phenocopies the OXPHOS mutants in this respect.

We performed quantitative real-time RT-PCR (qRT-PCR) analysis and confirmed that *AOX1*, *PEX11*, and *POT1* mRNAs were upregulated in WT in peroxisome proliferation conditions, as expected (*Figure 4E*). However, confirming a potential role of OXPHOS in SNF1 signaling, Δ*nugM* cells, like the Δ*sak1* cells, were unresponsive to oleate and methanol cultivation, and mRNA levels were not significant upregulated, relative to levels seen in WT cells, in any condition tested.

## Redox and ATP status of mutants affecting peroxisomal functions

To confirm our conclusions about the redox and ATP status in mitochondria, peroxisome, NADH shuttles, and SNF1 mutants during growth in peroxisome proliferation conditions, we developed cytosolic and peroxisome NAD$^+$/NADH sensors, and also measured oxygen consumption rates (OCR), which reflect mitochondrial ATP production (*Figure 5*). Due to the nature of the assays, we measured NAD$^+$/NADH ratios in oleate-grown cells and OCR in methanol growing cells. The OCR assay requires the presence of the carbon source in the medium because it measures energy production in live cells in real time and the viscosity of the oleate was incompatible with the instrument. The NAD$^+$/NADH ratio was measured as an endpoint intensity, so the presence of a carbon source was not needed during the assay and was not included during the acquisitions (see Materials and methods); however, the peroxisome sensors are transported to the peroxisomes by the PTS1 pathway, and mislocalization of the sensors will affect the assay. Oleate medium was chosen because the microscopy data from *Figure 1* indicated that BFP-PTS1 was transported more efficiently to peroxisomes in oleate medium than in methanol in mutants such as Δ*gpdA* Δ*mdhB*.

Several fluorescent protein-based NAD$^+$ biosensors were recently developed, providing insight into NAD$^+$ dynamics. We choose one of these sensors, SoNar, to measure NAD$^+$/NADH ratios (*Zhao et al., 2015*). SoNar was designed using a T-Rex NAD$^+$- and NADH-binding protein linked to cpYFP, and it shows a distinct fluorescence response to NADH and NAD$^+$. That is, the ratio of fluorescence intensities upon excitation at 485 and 420 nm decreases in the presence of high concentration of NADH, whereas the ratio increases with higher concentration of NAD$^+$. Like most of these sensors, SoNar is pH sensitive, but this was corrected by measuring SoNar and cpYFP fluorescence in parallel and normalization of the fluorescence intensity of SoNar with that of cpYFP.

We predicted that during growth in peroxisome proliferation conditions (oleate or methanol), dysfunctional peroxisomes, due to OXPHOS, SNF1, NADH- or acetyl-CoA-shuttling mutants or mutants lacking peroxisomal β-oxidation enzymes, such as Pot1, should not produce peroxisomal NADH, thereby decreasing the cytosolic pool of NADH, which should increase the cytosolic ratio of

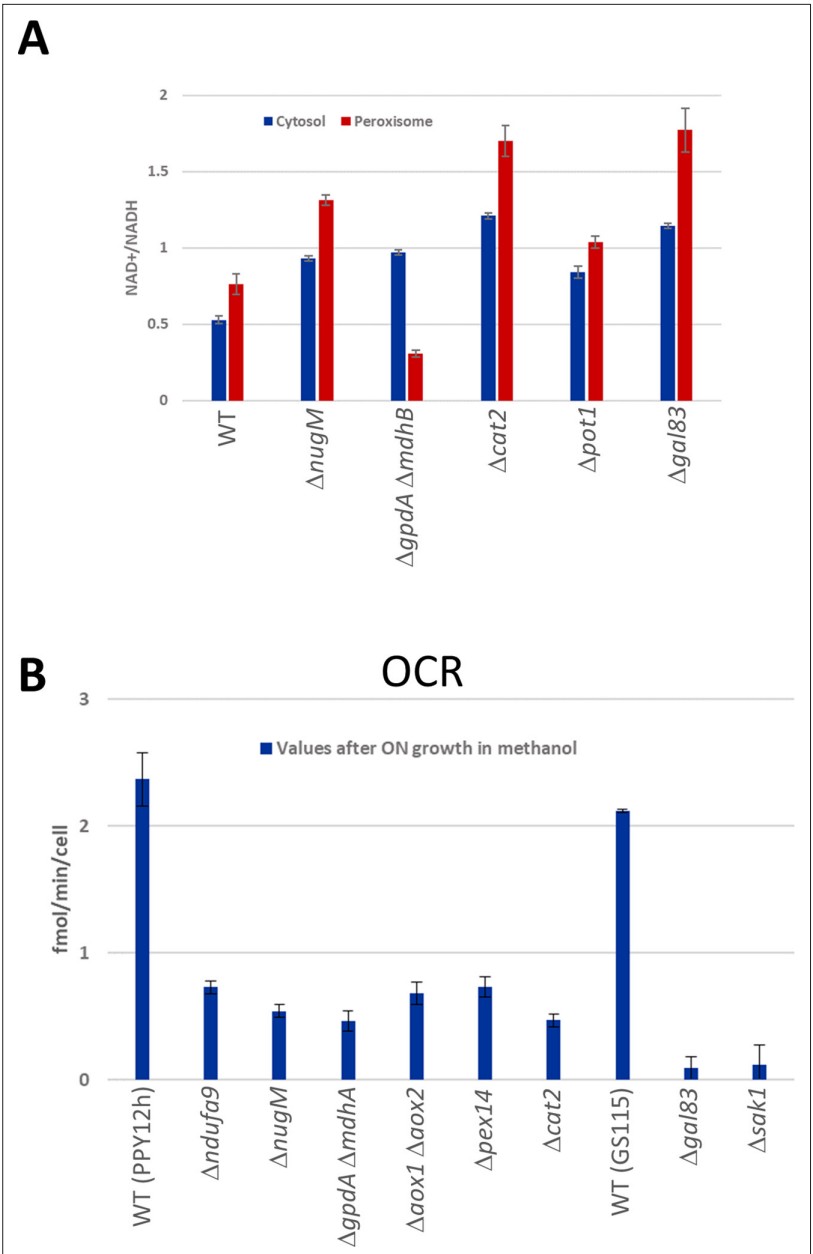

**Figure 5.** NAD$^+$/NADH ratios and oxygen consumption rates (OCRs). (**A**) NAD$^+$/NADH ratios obtained using SoNar sensors from oleate-grown cells (***Zhao et al., 2015***), as described in Materials and methods in wild-type and indicated mutant strains. (**B**) Cell number corrected OCR data from methanol-grown cells for wild-type and mutants from two different parental strains (PPY12h and GS115). Mutants generated from PPY12h: Δ*ndufa9*, Δ*nugM*, Δ*gpdA*Δ *mdhA*, Δ*aox1* Δ*aox2*, Δ*pex14*, and Δ*cat2*. Mutants generated from GS115: Δ*gal83* and Δ*sak1*. Each point in A and B corresponds to mean ± standard deviation (SD) of triplicate values. The p values (calculated using JMP statistical discovery version 16) were < 0.01 in Student paired *t*-tests between WT and mutant strains. Further details of p values and n numbers can be found in ***Source data 3***.

The online version of this article includes the following figure supplement(s) for figure 5:

**Figure supplement 1.** Extracellular acidification rates (ECARs).

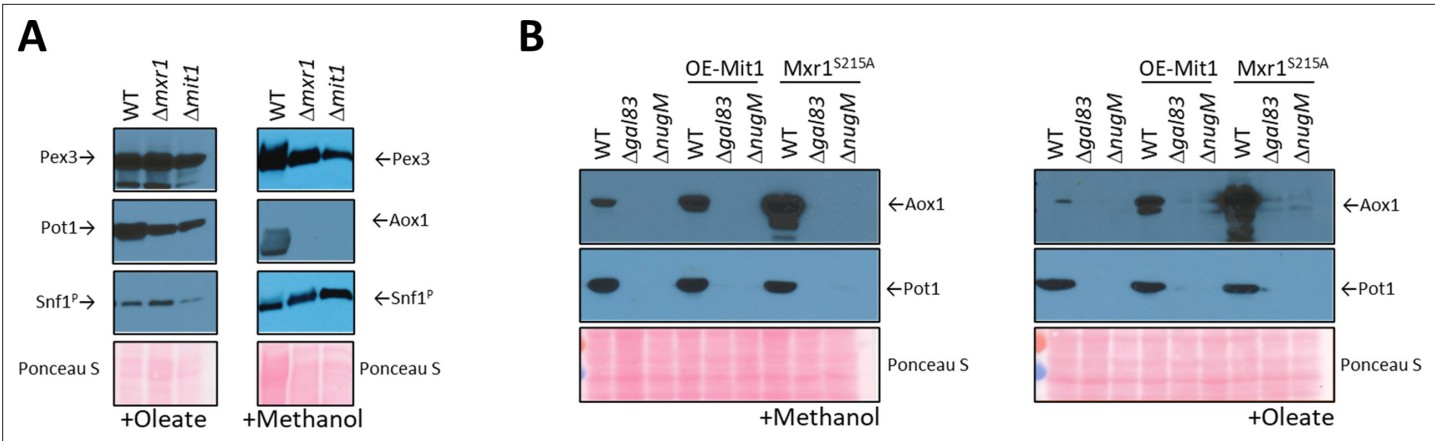

**Figure 6.** Analysis of transcriptional activators, Mxr1 and Mit1, of peroxisomal proteins. (**A**) Western blots of Pex3, phospho-Snf1, and Pot1 or Aox1 in WT, Δ*mxr1*, and Δ*mit1* mutant cells. (**B**) Western blots of Pot1 and Aox1 in WT, Δ*gal83*, and Δ*nugM* mutant cells, either expressing or not expressing the active form of Mxr1 (Mxr1^S215A). (**C**) Western blot of Pot1 in WT, Δ*gal83*, and Δ*nugM* mutant cells, without or with overexpression of Mit1 (OE-Mit1). Ponceau S staining was used as a loading control.

The online version of this article includes the following figure supplement(s) for figure 6:

**Figure supplement 1.** PKA inhibition or *HOG1* deletion did not rescue peroxisome proliferation defect of Δ*nugM* mutant cells.

NAD⁺/NADH compared to that in wild-type cells. Indeed, this was confirmed using SoNar (*Figure 5A*). Likewise, the ratios of NAD⁺/NADH in the peroxisome matrix were also expected to increase for those mutants for the same reasons, except for the NADH-shuttling mutant (Δ*gpdA* Δ*mdhB*) which should not increase because NADH cannot shuttle out of the peroxisomes, and this was also corroborated by this assay (*Figure 5A*).

In yeast, during growth on nonfermentable carbon sources, most if not all the oxygen, is consumed by OXPHOS to produce ATP. To assess the consequence of our mutants on ATP production during growth in peroxisome proliferation conditions we measured OCR using the Seahorse XF analyzer (*Figure 5B*). In addition, the Seahorse analyzer allows measurement of both OCR and the extracellular acidification rates (ECAR) of cells as indicators of glycolysis. The OCR values were significantly reduced in all the mutants tested compared to wild type, indicating ATP synthesis was seriously impaired. While this was highly expected for OXPHOS mutants, it confirmed the involvement of SNF1, NADH-shuttling, Cat2, and peroxisomal proteins in ATP synthesis during growth in methanol. ECAR, measured for the same samples, showed a significant activity boost after glucose supplementation, a good indication of the cell fitness of the sample set (*Figure 5—figure supplement 1*).

## OXPHOS defect in peroxisome proliferation is not rescued by active transcriptional activators of genes encoding peroxisomal proteins

As Snf1 is phosphorylated in cells with dysfunctional mitochondria, we hypothesized that OXPHOS mutations might affect peroxisomal gene expression downstream of Snf1 activation. We started by analyzing the role of Mit1 (Prm1 was excluded since it acts upstream of Mit1 in the same pathway *Wang et al., 2016a*), which functions during methanol induction and the SNF1-regulated factor, Mxr1, functioning primarily during glucose depression.

Contrary to our expectation that inactivation of one of these transcription factors might mimic the OXPHOS deficiency, we found that none of the deletions, Δ*mxr1* or Δ*mit1*, shared the phenotype of the Δ*nugM* strain (*Figures 4D and 6A*). During oleate induction, both Δ*mxr1* and Δ*mit1* strains showed only minor defects in Pot1 expression in comparison to WT cells, which was distinct from the behavior of the Δ*nugM* strain (*Figure 4D*). However, during growth in methanol, both transcription activators were essential for Aox1 expression, as described previously (*Wang et al., 2016a*; *Lin-Cereghino et al., 2006*; *Parua et al., 2012*; *Figure 6A*), confirming again the synergy of signals from the glucose derepression (SNF1 pathway) and from the methanol induction (Prm1 and Mit1) needed to activate the *AOX1* promoter.

The role of these transcription activators was studied further by their direct activation. Briefly, during glucose derepression, the constitutively expressed Mxr1 is activated by dephosphorylation at Ser 215 (*Parua et al., 2012*) and during methanol induction, Prm1 transmits the methanol signal to Mit1 by binding to the *MIT1* promoter, thus increasing expression of Mit1 (*Wang et al., 2016a*).

We used constitutively activated transcription activators to bypass any putative deficiencies in their activation in the OXPHOS mutants. We used Mxr1$^{S215A}$, inhibited PKA (PKA mutants are susceptible to 1NM-PP1 inhibition), which is the putative kinase for the inhibitory phosphorylation of the *S. cerevisiae* homolog of Mxr1 (Adr1), and overexpressed Mit1 (OE-Mit1) from the strong, constitutive *GAPDH* promoter (*Figure 6B*, *Figure 6—figure supplement 1*).

The plasmids expressing OE-Mit1, PKA mutants, and Mxr1$^{S215A}$ were transformed into WT, Δ*gal83*, and Δ*nugM* strains, respectively, and analyzed for the expression of Pot1 and Aox1 (*Figure 6B*, *Figure 6—figure supplement 1*). OE-Mit1 reduced the cell fitness independent of the background strain and cells grew slower in all carbon sources tested. Constitutive Mxr1 activation (Mxr1$^{S215A}$) did not affect the fitness of cells, but like OE-Mit1 or PKA inhibition, did not rescue the OXPHOS or SNF1 deficiencies (*Figure 6B*, *Figure 6—figure supplement 1*).

While Aox1 is strongly induced by methanol, the *AOX1* promoter is strictly repressed by other carbon sources such as glucose, glycerol, and ethanol (*Hartner and Glieder, 2006*). Recently, the *P. pastoris* MAP kinase, Hog1, was implicated in Aox1 repression under glycerol conditions (*Shen et al., 2016*) and in *S. cerevisiae*, Snf1 negatively regulates the activity of Hog1 (*Mizuno et al., 2015*). However, deletion of the *HOG1* gene in Δ*nugM* cells did not rescue the expression of Pot1 and Aox1 (*Figure 6—figure supplement 1*).

## Rescue of the OXPHOS defect by inactivation of transcriptional repressors of genes encoding peroxisomal proteins

The regulation of the transcription inhibitors, Mig1, Mig2, and Nrg1 by the SNF1 complex in *S. cerevisiae* has been clearly established, with Snf1 kinase directly phosphorylating Mig1 (*Treitel et al., 1998*) and Mig2 (*Chandrashekarappa et al., 2011*), or directly interacting with Nrg1 (*Vyas et al., 2001*), causing their release from the promoter of glucose-repressed genes, followed by their export to the cytosol.

In *P. pastoris*, the triple deletion strain (Δ*mig1* Δ*mig2* Δ*nrg1*) strongly derepressed Aox1 under glycerol-repression conditions, but not during glucose derepression, even when transcription activators (Mit1 and Prm1) were artificially activated (*Wang et al., 2017*). This makes sense because deletion of *MIG1* and *MIG2* significantly upregulated the binding of Mit1 at the *AOX1* promoter (*Shi et al., 2018*).

We used the already available Δ*mig1* Δ*mig2*Δ*nrg1* strain (*Wang et al., 2017*), expressing in it Pex3-GFP, and deleted the *NUGM* or *GAL83* (control) genes. Most of the peroxisome proliferation defects we observed in the OXPHOS and *gal83* mutants, such as peroxisome proliferation, as well as Pot1 and Aox1 expression, were rescued when cells were induced in either oleate or methanol (*Figure 7A, B*). Both quadruple mutants (Δ*mig1* Δ*mig2* Δ*nrg1* Δ*nugM* and Δ*mig1* Δ*mig2* Δ*nrg1* Δ*gal83*) behaved similarly in both media. Additionally, peroxisome proliferation was identical to that in WT and the triple mutant (Δ*mig1* Δ*mig2*Δ*nrg1*) strains (*Figure 7C, D*). We observed a reduction in Aox1 expression during oleate induction (glucose derepression) in the Δ*mig1* Δ*mig2* Δ*nrg1* Δ*nugM* strain, in comparison with the WT and the triple mutant strains, but this difference was also observed in Δ*mig1* Δ*mig2* Δ*nrg1* Δ*gal83* strain (*Figure 7A*), and was therefore considered insignificant. Thus, most of the peroxisome-associated defects caused the loss of mitochondrial ATP production can be rescued by inactivation of the three transcriptional repressors, closing the mechanistic loop involved in mitochondria–peroxisome interplay.

## Gal83 nuclear shuttling during peroxisome proliferation is inhibited in OXPHOS mutants

Transcriptional repressors and activators shuttle between the cytoplasm and the nucleus in a glucose-dependent manner. There is no doubt that the SNF1 complex needs to be localized in the nucleus to inactivate the repressors that occlude the promoters of glucose-repressed genes, but there is little to no evidence for Mxr1/Adr1 activation in the nucleus. Instead, if Mxr1/Adr1 indeed can be activated within the cytosol, a deficient nuclear enrichment of SNF1 in the OXPHOS mutant during glucose

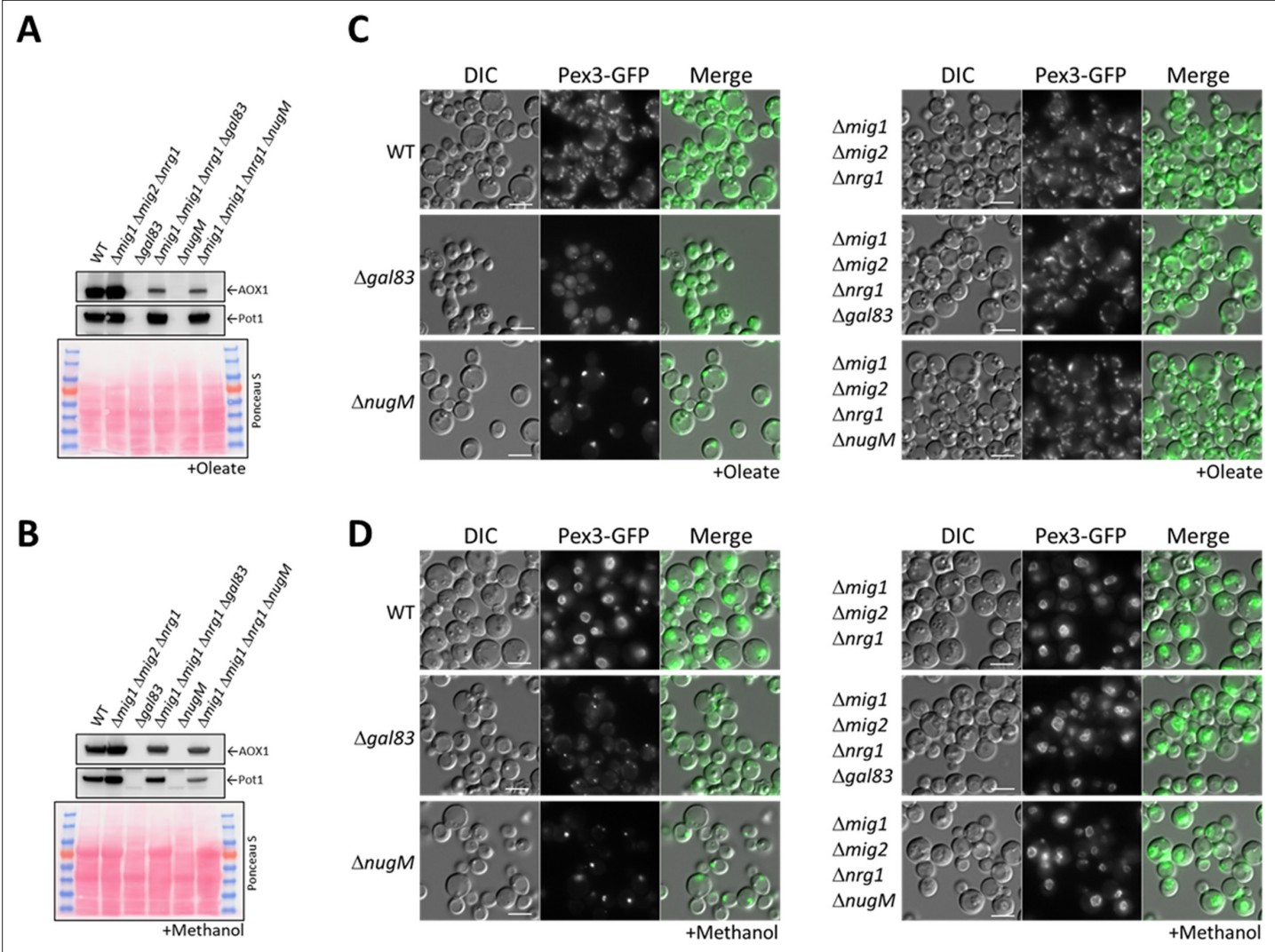

**Figure 7.** Deletion of transcriptional repressors regulated by SNF1 complex signaling rescues Δ*nugM* and Δ*gal83* mutant cells. (**A, B**) Western blot of Aox1 and Pot1 in WT, Δ*nugM* and Δ*gal83* mutant cells, with and without deletions of genes (*MIG1*, *MIG2*, and *NRG1*) encoding the transcriptional repressors regulated by SNF1 signaling. Ponceau S staining was used as a loading control. (**C, D**) Fluorescence microscopy of WT, Δ*gal83*, and Δ*nugM* mutant cells expressing Pex3-GFP, with and without deletions of genes (*MIG1*, *MIG2*, and *NRG1*) encoding the transcriptional repressors. Cells were grown in oleate and methanol for 16 hr, respectively. Bar: 5 μm.

derepression could explain the rescue of peroxisome biogenesis defects in the Δ*nugM* mutant by deletion of the repressors. In *S. cerevisiae*, SNF1 complex localization is regulated by the β-subunits (Gal83, Sip1, and Sip2), with Gal83 being the subunit responsible for the nuclear localization of the complex (referred to hereafter as SNF1–Gal83).

We tested for effects of the of Δ*nugM* mutant on Gal83 nuclear localization by using a Gal83-GFP fusion expressed from the native *GAL83* promoter. WT and mutant cells were grown on abundant glucose, and the nuclear enrichment of Gal83-GFP was stimulated by shifting the cells to oleate for 30 min or methanol for 2 hr in the presence of leptomycin B (LMB, a potent and specific Crm1 nuclear export inhibitor, required for Gal83 nuclear export in *S. cerevisiae*; *Hedbacker and Carlson, 2006*). As expected, Gal83-GFP was excluded from the nuclei of glucose-grown WT, Δ*sak1*, and Δ*nugM* cells (*Figure 8*, *Figure 8—figure supplement 1*). Upon the shift to methanol or oleate, Gal83-GFP was enriched in the nucleus of the WT (labeled with the perinuclear ER marker, Sec61-mCherry) and excluded from the nucleus of the Δ*sak1* mutant, as anticipated. Interestingly, Δ*nugM* cells, like Δ*sak1* cells, showed no nuclear enrichment of Gal83-GFP. Together with the Snf1 activation assay (*Figure 4D*),

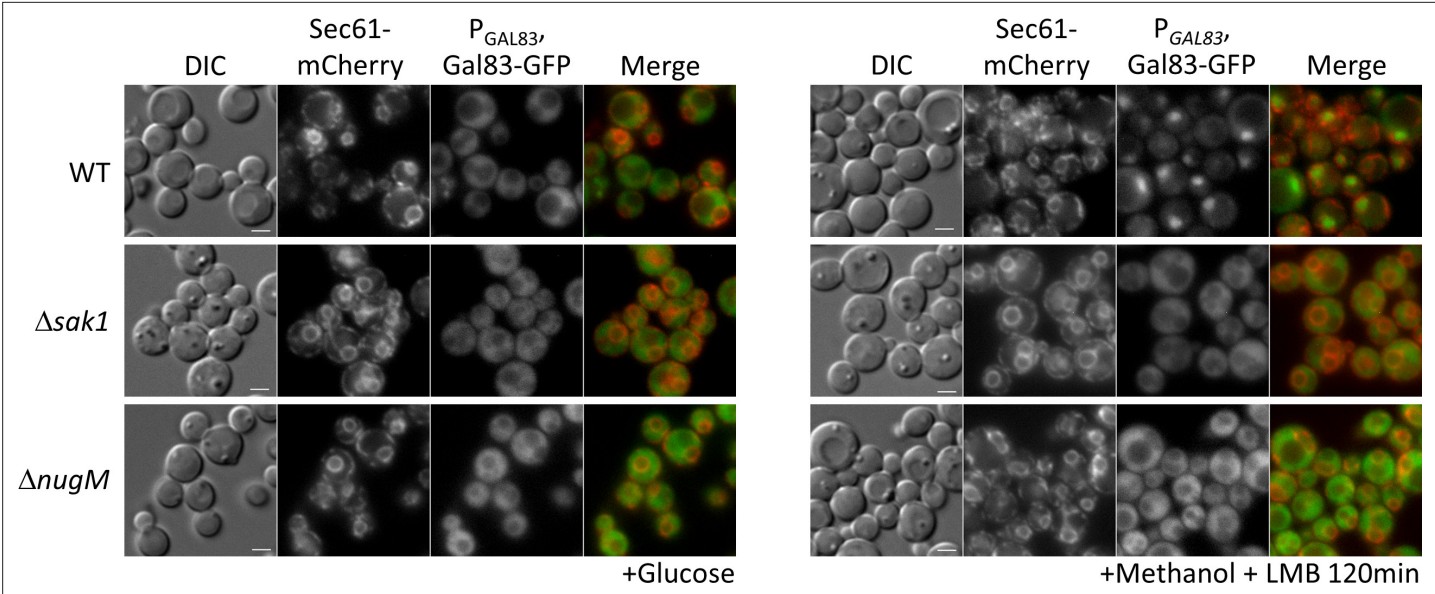

**Figure 8.** Gal83 nuclear localization during methanol adaptation is inhibited in Δ*nugM* and Δ*sak1* mutant cells. Fluorescence microscopy of WT, Δ*sak1*, and Δ*nugm* mutant cells expressing Gal83-GFP driven by the *GAL83* promoter in the presence of 200 ng/ml leptomycin B (LMB) and Sec71-mCherry as perinuclear ER marker. Bar: 5 µm.

The online version of this article includes the following figure supplement(s) for figure 8:

**Figure supplement 1.** Gal83 nuclear localization during oleate adaptation is inhibited in Δ*nugM* mutant cells.

these results strongly suggest that the OXPHOS mutants are impaired in the nuclear localization of the SNF1–Gal83 complex independent of the catalytic activation of Snf1 kinase.

## Peroxisome proliferation in Δ*pex14* cells can be induced by growth in lactate medium

The aberrant regulation of the SNF1 signaling pathway in cells with dysfunctional mitochondria during glucose derepression, and the lack of a normal peroxisome proliferation in NADH-shuttling mutants during growth in oleate and methanol suggest the presence of a feedback loop between both organelles which goes beyond the initial activation of the SNF1 signaling pathway by glucose derepression. This loop refers to the influence of peroxisomal NADH on mitochondrial ATP production, which in turn feeds back to influence peroxisome biogenesis, division, and proliferation.

We tested the feedback loop hypothesis using a strain lacking Pex14, a key member of the peroxisome docking complex and essential for import of peroxisomal matrix proteins (*Farré et al., 2019*). Although normal peroxisomes are absent in cells lacking Pex14, abnormal vesicles containing PMPs but lacking matrix proteins, are observed as peroxisomal remnants. These remnants can be visualized with Pex3-GFP and when we compared Δ*pex14* mutant to WT cells after cultivation in oleate medium, we observed a significantly lower number of peroxisomal structures (*Figure 9*). We investigated if an inactive OXPHOS (due to the lack of NADH production by dysfunctional peroxisomes) is the reason for the low number of remnants. We noticed after 4 hr of cultivation in L-lactate (lactate) medium (nonfermentable carbon source), that WT cells showed higher levels of Pex11, Aox1, and Pot1 proteins compared to glucose depletion or to cultivation in methanol or oleate (*Figure 4D*), which could be a consequence of the lactate metabolism by the mitochondria. Therefore, we used the lactate medium to overcome the low levels of NADH in Δ*pex14* cells, and as we hypothesized, the numbers of peroxisome structures increased in Δ*pex14* cells and were comparable to those in WT cells cultivated in lactate medium.

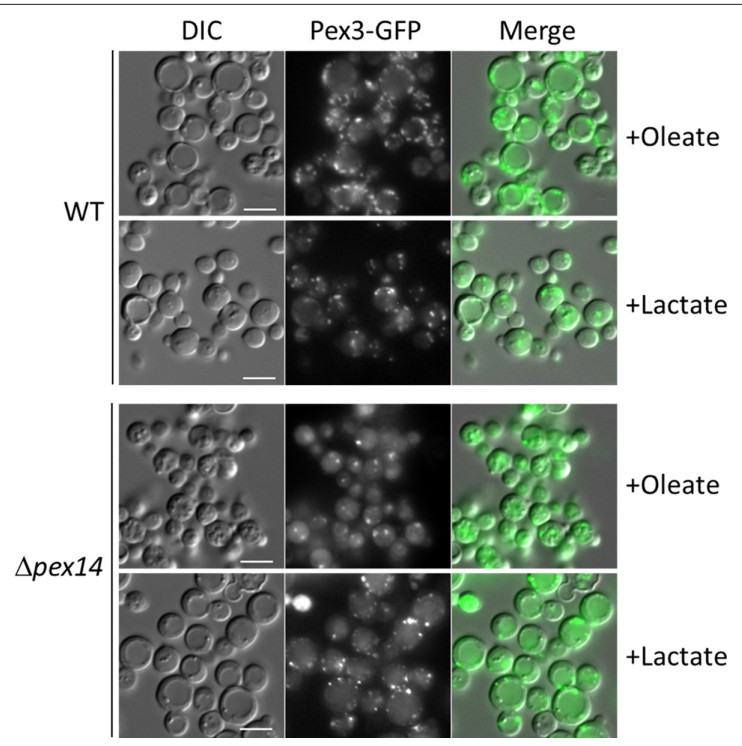

**Figure 9.** Cultivation of Δ*pex14* mutant cells in the respiratory medium, lactate, induces peroxisome proliferation. Fluorescence microscopy of WT, Δ*pex14* mutant cells expressing Pex3-GFP in different media for 8 hr. Bar: 5 μm.

## Discussion

Organelles are often viewed as individual entities with defined composition and organization that endow them with specialized functions. However, it is now clear that intracellular membrane compartments engage in extensive communication, either indirectly, or directly through membrane contacts (*Fransen et al., 2017*; *Castro et al., 2018*; *Chu et al., 2015*; *Schuldiner and Zalckvar, 2017*; *Shai et al., 2016*). Mitochondrial dysfunction impacts several other organelles and their biogenesis, including peroxisomes (*Diogo et al., 2018*), but the underlying mechanisms have not been elucidated. We have uncovered here the mechanisms involved in one mode of interorganelle communication and interplay in *P. pastoris* where the mitochondria and peroxisomes sense the metabolic status of the cell, influence each other's metabolism, in concert with cytosolic and nuclear involvement, and regulate peroxisome proliferation and division as needed.

### Interorganellar communication and interplay between peroxisomes and mitochondria

In the mitochondrial OXPHOS mutants (Δ*nugM*, Δ*ndufa9*, and Δ*cyt1*), and upon using DNP, peroxisome proliferation, division, and biogenesis of several peroxisome-associated proteins, but not peroxisomal matrix protein import (*Figure 2*), are affected in cells grown in either methanol or oleate, illustrating the mitochondrial involvement in multiple peroxisomal processes. We extended earlier reports that intermediates of peroxisome metabolism, or the absence of peroxisomal enzymes, regulate the maturation and fission of the organelle (*Nguyen et al., 2006*; *Espeel et al., 1997*) by several additional observations. The lack of peroxisomal thiolase (Pot1) affected peroxisome proliferation and division exclusively in oleate. Similarly, the absence of Aox1 and Aox2, involved in the methanol utilization pathway, affected peroxisome proliferation and division only in cells grown in methanol medium (*Figure 1—figure supplement 1*). Finally, the lack of Pex5, responsible for the import of PTS1-containing enzymes (including some involved in the β-oxidation and methanol metabolism pathways), impaired peroxisome proliferation and division in both media (*Figure 1—figure supplement 1*). Additionally, blocking the entry of NADH into mitochondria using double deletion strains, Δ*mdhA*

Δgpd1 and ΔmdhB Δgpd1 cells, yielded a phenotype like that seen in the OXPHOS mutants, wherein most of the cells contained only a single, import-competent peroxisome (*Figure 1C*). These results are consistent with the role of NADH-shuttling proteins, responsible for shuttling NADH between both peroxisomes and mitochondria via the cytosol (*Figure 5A*), in the interorganellar interplay between these organelles, and also points to a role for the cytosol.

## Feedback loop between peroxisomes and mitochondria senses cellular metabolic status

Our results in *P. pastoris* during glucose derepression indicate that peroxisome proliferation, division and biogenesis rely on a functional OXPHOS. Moreover, as methylotrophic yeast rely exclusively on peroxisome metabolism to feed into the mitochondrial OXPHOS for energy production during growth in FA or methanol (*Kurihara et al., 1992*; *Tanaka et al., 1982*), this feedback loop between the two organelles might contribute to sensing the cell's metabolic status during cultivation in these carbon sources (*Figure 10*). Key elements of this feedback loop demonstrated here include the requirement of NADH produced by mitochondria, peroxisomes and NADH shuttles, mitochondrial ATP production, SNF1 activation including Gal83 nuclear translocation and removal of repression by Mig1, Mig2, and Nrg1 for peroxisome biogenesis, division and proliferation. We further validated this feedback loop when we induced the peroxisome proliferation in Δpex14 cells, which cannot import peroxisomal matrix proteins or produce NADH, by cultivation in lactate (*Figure 9*), which, in yeast, can only be metabolized by mitochondria. Any externally added lactate will enter mitochondria via a putative lactate/proton symporter and will then be oxidized to pyruvate with a reduction of cytochrome C to generate ATP (*Passarella et al., 2008*). In Δpex14 cells, cultivation in lactate activated OXPHOS by mitochondrial metabolism and overrode the need for NADH produced by peroxisomal metabolism, boosting peroxisome proliferation (*Figure 9*). This also shows clearly that the molecule triggering peroxisome proliferation does not emanate from within peroxisomes, which are dysfunctional in Δpex14 cells.

## The OXPHOS effect on peroxisome phenotypes in mediated by the absence of SNF1–Gal83 translocation to the nucleus

The requirement of ATP we observed for peroxisome biogenesis, division, and proliferation is likely mediated through the SNF1 complex that responds to altered cellular AMP/ATP ratios. Our data show that the phenotype of the OXPHOS mutants was due to the inability of the mutants to produce ATP resulting in the lack of nuclear translocation of the SNF1–Gal83 complex (*Figure 8*) and a defect in the transcriptional induction of peroxisome division (Pex11) and peroxisomal matrix proteins (e.g. Pot1 and Aox1) in the mutants grown in methanol or oleate (*Figure 4D*). This reiterates a nuclear involvement in the activation of peroxisome proliferation.

We present several lines of evidence showing that the newly discovered role played by OXPHOS in nuclear enrichment of the SNF1–Gal83 complex is separable from its role in catalytic activation of the kinase, and it rather reflects a role in regulating the nuclear relocalization of Gal83 itself. First, the ΔnugM mutation alone does not affect Thr 210 phosphorylation and activation of Snf1 kinase (*Figure 4D*). Second, Gal83 fails to enrich in the nucleus of the ΔnugM mutant, under conditions where this relocation is evident in WT cells (*Figure 8*). The pathway leading to Gal83 import and the proteins involved in this process are not known. Gal83 may undergo a posttranslational modification upon glucose derepression that could promote a change in cellular localization. In *S. cerevisiae*, Gal83 phosphorylation sites have been identified in high-throughput studies (*Lanz et al., 2021*), but the effect of such modifications is not known. Some evidence suggests a role for the mitochondrial voltage-dependent anion channel (VDAC) protein, Por1, promoting Gal83 nuclear localization by a mechanism that is distinct from Snf1 activation during growth in glycerol/ethanol (*Shevade et al., 2018*), and for the nuclear export receptor, Crm1, in the nuclear exclusion of Gal83 during growth on abundant glucose (*Hedbacker and Carlson, 2006*). *P. pastoris* OXPHOS mutants share phenotypes of ΔScpor1 mutant, such as normal Snf1 activation by glucose derepression and the absence of induction of genes regulated by SNF1 signaling (*Shevade et al., 2018*). However, the deletion of the only *POR1* homolog, or overexpression of Por1 in *P. pastoris*, did not produce the same phenotypes observed in the OXPHOS mutants and Gal83 localization was unaffected by Por1. Corroborating the role of Gal83 was also the finding the Δgal83 cells phenocopied the OXPHOS mutants (*Figure 4C*).

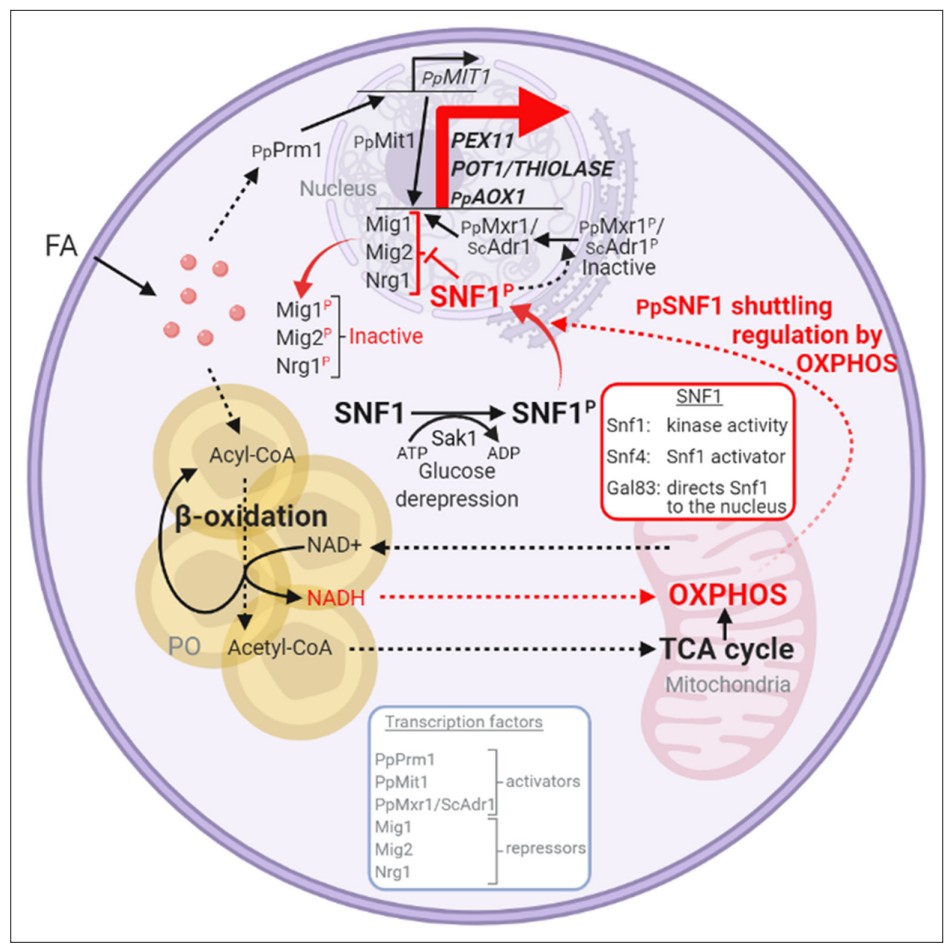

**Figure 10.** Interorganellar communication and signaling pathways in peroxisome proliferation, division, and matrix protein biogenesis. Feedback loop between peroxisome and mitochondria is shown in red. Fatty acids (FA) uptake and its β-oxidation produce NADH equivalents and acetyl-CoA. However, the peroxisome membrane is impermeable to large hydrophilic solutes, including $NAD^+$, NADH, $NADP^+$, and NADPH, as well as ATP and acylated or unacylated coenzyme A (CoA) (**Wanders et al., 2020**). Consequently, NADH-shuttling proteins, working together in the peroxisomes, cytosol, and mitochondria, deliver NADH to mitochondria to feed oxidative phosphorylation (OXPHOS; **Farré et al., 2021**). Acetyl-CoA produced in peroxisomes is delivered to mitochondria via acetyl-carnitine produced in peroxisomes, and mitochondria use the TCA cycle and OXPHOS for full oxidation to $CO_2$ and $H_2O$ (**Wanders et al., 2020**). The SNF1/AMP-activated protein kinase (AMPK) complex, which is sensitive to the cellular AMP:ATP ratio, maintains the balance between ATP production and consumption (**Kayikci and Nielsen, 2015**; **Kim et al., 2013**; **Yan et al., 2018**). Glucose deprivation, which reduces ATP production, activates Snf1 (by phosphorylation) via the action of the Sak1 kinase, and in *P. pastoris*, the nuclear translocation of the SNF1–Gal83 complex requires OXPHOS and ATP production. Peroxisome-associated proteins, such as the division protein, Pex11, and the matrix proteins, Pot1 and Aox1, are regulated negatively by transcriptional repressors, that compete with transcriptional activators, such as Mit1 and Mxr1 (equivalent to ScAdr1). Snf1 activation in the cytosol and SNF1–Gal83 entry into the nucleus removes, by phosphorylation of the appropriate proteins, the repression of expression of the peroxisome-associated proteins, while also activating the transcriptional activators. This simultaneous action of SNF1–Gal83 turns on the biogenesis of peroxisome-associated proteins, peroxisome proliferation, as well as division. In the Δ*nugM*, Δ*ndufa9*, Δ*gal83*, Δ*pot1*, Δ*aox1* Δ*aox2*, and Δ*pex5* mutants of *P. pastoris*, or in the presence of 2,4-dinitrophenol (DNP), peroxisome proliferation, division, and the biogenesis of certain peroxisome-associated proteins in compromised. FA, fatty acids; PO, peroxisome; P, phosphorylation.

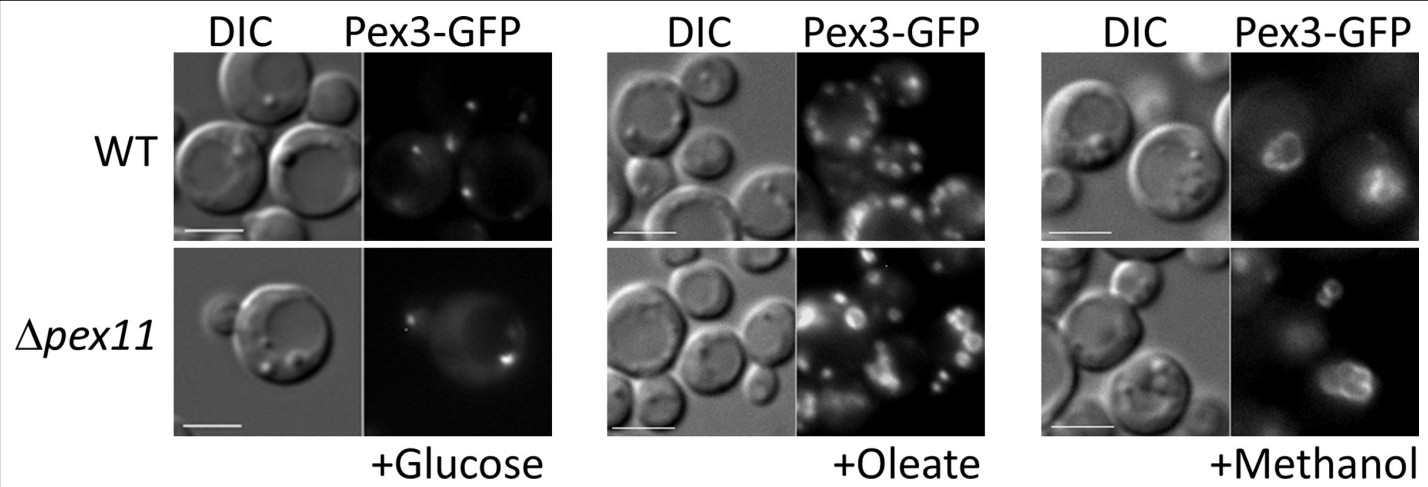

**Figure 11.** The Δ*pex11* cells of *P pastoris* do not phenocopy the oxidative phosphorylation (OXPHOS) mutant, and can proliferate peroxisomes in oleate and methanol. Peroxisomes are labeled with Pex3-GFP. Bar: 5 μm.

## The downstream targets of the SNF1–Gal83 signaling explain the peroxisome-related phenotypes in the OXPHOS mutants

Among the proteins whose expression is affected transcriptionally in the OXPHOS mutants, is Pex11. The Pex11 family of proteins is conserved in yeasts, plants, and mammals, and orchestrates peroxisome division (*Koch and Brocard, 2012*; *Koch et al., 2010*; *Aung et al., 2010*; *Tam et al., 2003*; *Thoms and Erdmann, 2005*). Its absence blocks peroxisome division in yeast, but importantly, not its de novo proliferation (*Huber et al., 2012*), which is seen quite clearly in *P. pastoris* where Δ*pex11* mutants grown in oleate have only 1–2 large peroxisomes, in comparison to WT cells that have about 7 peroxisomes/cell (*Joshi et al., 2012*). However, in methanol there is indeed peroxisome proliferation and cells have multiple peroxisomes in cells lacking either Pex11 or its downstream peroxisome division component, Fis1 (unpublished data cited in *Joshi et al., 2012*). Additionally, the *P. pastoris* pex11 mutant does not phenocopy the OXPHOS mutant (*Figure 11*). In both *S. cerevisiae* and *P. pastoris*, Pex11 is induced significantly (*Karpichev and Small, 1998*) upon switch from YPD to oleate (*Joshi et al., 2012*). In *P. pastoris*, Pex11 is fully repressed upon growth of cells in YPD, whereas in *S. cerevisiae*, Pex11 is expressed at a low level even in YPD (*Joshi et al., 2012*). The absence of induction of Pex11 in oleate-grown cells (*Figure 4A, D*) is likely the underlying cause for the lack of peroxisome division in the OXPHOS mutants, but would not explain the additional absence of peroxisome proliferation.

The biogenesis defects we observed for peroxisome-associated proteins in the OXPHOS mutants are likely caused by the lack of induction of peroxisomal matrix proteins, such as Pot1 and Aox1 (*Figure 4C–E*), and their corresponding RNAs (*Figure 4D*). Further corroboration of this conclusion comes from our data that the absence of several peroxisomal matrix proteins (Pot1, Aox1, and Aox2) by deletion of their corresponding genes, or by prevention of their import into peroxisomes (*PEX5* gene deletion) also affects peroxisome proliferation, as seen for the OXPHOS mutants (*Figure 1—figure supplement 1*).

Finally, the rescue of the phenotype of the OXPHOS mutant by deletion of the genes encoding the repressors Mig1, Mig2, and Nrg1, extends not only to the induction of *PEX11*, *POT1*, and *AOX1* genes (*Figure 7A, B*), but also to the reversion of the defect in peroxisome proliferation (*Figure 7C, D*), making it very likely that the missing factor required for peroxisome proliferation in the OXPHOS mutants must be a direct or indirect target of these transcriptional repressors, but as argued earlier, based on the peroxisome proliferation seen in Δ*pex14* cells in lactate, this is unlikely to involve a peroxisomal metabolite. In contrast, peroxisome division can be influenced by peroxisomal metabolites (*Rinaldi et al., 2016*). Identification of this proliferation factor remains an important future priority.

We rescued the peroxisome proliferation, peroxisomal protein biogenesis, and division defects in the OXPHOS and Δ*gal83* mutants, by the simultaneous deletion of three genes encoding the

transcriptional repressors Mig1, Mig2, and Nrg1 (*Figure 7*), which repress genes required for peroxisome biogenesis and function (*Shi et al., 2018*; *Wang et al., 2016b*). Deletions of the genes for the transcriptional activators, Mit1 and Mxr1, known to activate peroxisome-related genes (*Wang et al., 2016a*; *Lin-Cereghino et al., 2006*), individually, did not yield the same phenotype as that of the OXPHOS mutants (*Figure 6A*), suggesting that these are not directly impaired in the OXPHOS mutants. Additionally, neither overexpression of Mit1, nor artificial activation of Mxr1, rescued the similar phenotypes of the Δ*nugM* and Δ*gal83* mutants (*Figure 6B*). Therefore, most of the peroxisome-related phenotypes of the OXPHOS mutants can be explained by the lack of relief of repression of several genes by Mig1, Mig2, and Nrg1, thereby providing a molecular explanation for how the ATP generated by the mitochondria impacts peroxisome proliferation, division, and biogenesis.

## Working model for interorganellar control of peroxisome dynamics

We present a working model illustrating the interorganellar transactions, communications, and signaling that must occur in *P. pastoris* between peroxisomes, cytosol, mitochondria, and the nucleus (see *Figure 10* legend for details) for peroxisome proliferation, division, and biogenesis of peroxisome-associated proteins. FA uptake and its -oxidation produce NADH equivalents and acetyl-CoA. However, the peroxisome membrane is impermeable to large hydrophilic solutes, including $NAD^+$, NADH, $NADP^+$, and NADPH, as well as ATP, and either acylated or unacylated coenzyme A (CoA) (*Wanders et al., 2020*). Consequently, NADH-shuttling proteins, working together in the peroxisomes, cytosol, and mitochondria, allow the delivery of NADH to mitochondria to feed OXPHOS (*Farré et al., 2021*). Acetyl-CoA produced in peroxisomes is delivered to mitochondria via acetyl-carnitine produced in peroxisomes, and mitochondria use the TCA cycle and OXPHOS system for full oxidation to $CO_2$ and $H_2O$ (*Wanders et al., 2020*).

The SNF1/AMPK complex, which is sensitive to the cellular AMP:ATP ratio, maintains the balance between ATP production and consumption in all eukaryotic cells (*Kayikci and Nielsen, 2015*; *Kim et al., 2013*; *Yan et al., 2018*). Glucose deprivation, which reduces ATP production, activates Snf1 (by phosphorylation) via the action of the Sak1 kinase, and in *P. pastoris*, the nuclear translocation of the SNF1–Gal83 complex requires OXPHOS and ATP production, as shown here. Peroxisome-associated proteins, such as the division protein, Pex11, and the matrix proteins, Pot1 and Aox1, are regulated negatively by transcriptional repressors, that compete with transcriptional activators, such as Mit1 and Mxr1 (equivalent to ScAdr1). Snf1 activation in the cytosol and SNF1–Gal83 entry into the nucleus removes, by phosphorylation of the appropriate proteins, the repression of expression of the peroxisome-associated proteins, while also activating the transcriptional activators. This simultaneous action of SNF1–Gal83 turns on the biogenesis of peroxisome-associated proteins, peroxisome proliferation, as well as division. In the Δ*nugM*, Δ*ndufa9*, Δ*gal83*, Δ*pot1*, Δ*aox1 aox2*, and Δ*pex5* mutants of *P. pastoris*, or in the presence of the mitochondrial uncoupler, DNP, peroxisome proliferation, division, and the biogenesis of certain peroxisome-associated proteins in compromised.

In conclusion, since ATP production by mitochondrial OXPHOS is impaired in many human diseases involving over 150 genes (*Liu et al., 2021*), including Parkinson's disease and schizophrenia (*Bergman and Ben-Shachar, 2016*; *López-Gallardo et al., 2011*; *Zhu and Wang, 2017*), as well as during aging (*Olgun and Akman, 2007*) and neurodegeneration (*Koopman et al., 2013*), the results presented here suggest a mechanistic link to peroxisomal dysfunction in human mitochondrial disorders. Further explorations of this yeast model would provide important insights regarding interorganellar communication, interplay, and dynamics, while also shedding light on human disease.

## Materials and methods

*Methods availability statement* – Strains and plasmids are described in *Supplementary file 1*. Those constructed for the purpose of this study can be requested from the Subramani Lab, following UC San Diego's MTA guidelines (https://blink.ucsd.edu/research/conducting-research/mta/index.html).

## Media and reagents used to grow strains

*P. pastoris* strains were prototrophic without requiring amino acid supplements or yeast extract and were cultivated in minimal media containing different carbon sources. Supplementation of media with amino acids did not change the results of this study. YPD (2% glucose, 2% bacto-peptone, 1%

yeast extract), glucose medium (2× YNB [YNB: 0.17% yeast nitrogen base without amino acids and ammonium sulfate, 0.5% ammonium sulfate], 0.04 mg/l biotin, 2% dextrose), oleate medium (2× YNB, 0.04 mg/l biotin, 0.02% Tween-40, 0.2% oleate), and methanol medium (2× YNB, 0.04 mg/l biotin, 1% methanol). Amino acids were supplemented when indicated using 0.79 g/l complete synthetic medium of amino acids and supplements (CSM; #1001, Sunrise Science Products, USA). DNP (#D198501, Sigma-Aldrich, USA) was dissolved in methanol, used at a final concentration of 0.25 mM and added after switching cells from glucose to methanol (1%). LMB solution (#L2913, Sigma-Aldrich, USA) was added to the methanol medium at final concentration of 200 ng/ml.

## Plasmid constructions

Plasmids were constructed by Gibson Assembly for which primers were designed using NEBuilder (https://nebuilder.neb.com/#!/). DNA was amplified from WT genomic DNA by PCR using Advantage 2 Polymerase (#639202, Takara Bio, USA). Plasmid backbones were double digested with the necessary restriction enzymes and purified using the Qiagen DNA purification kit (#28704 × 4, QIAGEN GmbH, Germany). DNA inserts were cloned into the digested vectors using the NEBuilder Hifi DNA Assembly Mastermix (#E2621L, New England Biolabs).

## Yeast strain constructions

Plasmids were linearized with the appropriate restriction enzyme before transformation of competent yeast cells by electroporation (*Cregg and Russell, 1998*). The transformed cells were plated on YPD plates with the appropriate selection markers and incubated at 30°C for a few days. Colonies were screened by western blot or fluorescence microscopy.

## Fluorescence microscopy

Cells were grown in YPD at 30°C until exponential phase (1–2 $OD_{600}$/ml), washed twice with sterile water, and then transferred to glucose or peroxisome proliferation (methanol or oleate) media. With the goal of keeping cells at exponential phase, different starting $OD_{600}$ were used for different strains, depending on the strain's cell doubling time in the respective media. Cells were grown in indicated media in 250 ml flasks at 30°C and shaken at 250 rpm. Cells were pelleted, washed twice with sterile water and 1.5 µl of cells were mixed with 1% low melting point agarose and placed on a glass slide with a cover slip and imaged using ×63 or ×100 magnification on a Carl Zeiss Axioskop fluorescence microscope. Images were taken on an AxioCam HR digital camera, no digital gain was used, exposition for peroxisome markers was kept constant. Images processed on AxioVision software are representative results from experiments conducted at least in triplicate.

## Biochemical studies

The same conditions described for fluorescence microscopy assays were used to grow cells. Five $OD_{600}$ of cells was collected at different times as described in the figures, trichloroacetic acid precipitated and analyzed by sodium dodecyl sulfate–polyacrylamide gel electrophoresis and then by western blot. The commercial antibodies used are as follows: Anti-HA (#11666606002, Roche, Germany), Anti-Snf1 (#100G7E, Cell Signaling Technology, USA), Anti-Rabbit HRP (#172-1019, BioRad, USA), Anti-Mouse HRP (#1706516, BioRad, USA), and Anti-Rat HRP (#ab97057, Abcam, USA). Additionally, the following antibodies were generated in-house: Anti-Pex2, Anti-Pex3, Anti-Pot1, and Anti-Aox1.

*qRT-PCR* – RNA extraction was performed using Trizol Reagent (#15596026, Invitrogen, USA) following the manufacturer's instructions with some modifications. Twenty $OD_{600}$ of frozen cells, grown in the conditions described, were resuspended with 1 ml of Trizol and 250 µl of acid-washed glass beads (425–600 µm). The suspension was vortexed for 30 s, followed by chilling on ice for 30 s. This step was repeated three times. The extraction was completed according to the protocol described by the kit manufacturer and was followed by DNAse I treatment (#18068015, Life technologies, Carlsbad, CA, USA). The RNA quantity and quality were determined using the 260/280 nm ratio of approximately 1.8–2.0 for good RNA quality. The RNA samples were stored at (add symbol 80° C). cDNA was synthesized from 1 µg of total RNA using an Invitrogen two-step kit with SuperScript III (#18080051, Invitrogen, Carlsbad, CA, USA) as the reverse transcriptase (RT) enzyme with random hexamer. This was followed by RNAse treatment. The cDNAs were stored at add symbol 80°C. One µl of cDNA (equivalent to 80 ng of total RNA), 300 nM of primers

listed in *Supplementary file 2* and PowerUp SYBR Green master mix (#A25741, Applied Biosystems, USA) were used. Values for each target gene were normalized using 18S rRNA and WT in glucose condition as reference. Expression values were calculated using the $2^{-\Delta\Delta CT}$ method (*Livak and Schmittgen, 2001*).

## Metabolic analysis – Biolog Mitoplate assay

Mitochondrial metabolic activity was measured in digitonin-permeabilized cells using the PM1 Micro-Plate (Biolog #12111, Hayward, CA, USA) following the manufacturer's instructions (protocol document dated February 6, 2019 using cell preparation protocol Option 1). Briefly, cells were grown in lactate medium, washed and resuspended in high lactose osmotic stabilizing solution (YMAS), containing digitonin (#D-180-250, Gold Biotechnology, USA). After 1 hr, the permeabilized cells were seeded into 96-well PM1 MicroPlates containing different potential energy substrates. The colorimetric assay was initiated by adding Redox Dye Mix MC (Biolog #74353). The PM1 MicroPlate was then loaded into the OmniLog PM-M system (Biolog, Hayward, CA, USA) for kinetic reading at 30°C. The MitoPlates were read for 24 hr at 15 min intervals at OD590. OmniLog rate values were calculated using Data Analysis 1.7 software before being transferred to a Microsoft Excel sheet for reformatting. The software calculates the maximal rate change from the gradient of the reading output during the 24 hr time period. These data are presented as raw OD590 (A.U.).

## Seahorse

OCR and extracellular acidification rate were determined in a Seahorse XF96 analyzer. In brief, for the OCR and ECAR analysis, between $5–10 \times 10^4$ cells, coming from 15 to 16 hr methanol cultures, were added to a poly-lysine coated Seahorse XF96 PDL Cell Culture Microplate (#103730-100, Agilent, USA) in 50 µl of assay medium (XF DMEM medium pH 7.4 with 5 mM HEPES without phenol red, sodium bicarbonate, glucose, L-glutamine and sodium pyruvate (#103575, Agilent, USA) supplemented with 6 mM glutamine solution (#103579, Agilent, USA) and 1% methanol). The plate was centrifuged at $3000 \times g$ for 20 min with gentle acceleration and deceleration. Then 130 µl of assay medium were added to each well and incubated at 30°C in a non-CO$_2$ incubator for 1 hr. Immediately following completion of the incubation, OCR and ECAR was measured three times every 10 min and an additional three times every 30 min after the injection of glucose (2% final concentration; #103577-100, Agilent, USA). All samples were analyzed in triplicate, normalized by cell number and the third stable reading of every condition was plotted in respective graphs.

## SoNar

NAD$^+$/NADH ratio was measured using the genetically encoded sensor, SoNar and its control cpYFP. Cells were pregrown in oleate medium for 8 hr, washed twice and resuspended in 2× YNB supplemented with 0.04 mg/l biotin. Between $2–3 \times 10^7$ cells in 200 µl were seeded in a 96 well black/clear plate (#353219, BD Falcon, USA) and end point fluorescence intensities were measured in a NOVOstar Microplate Reader (BMG LABTECH, USA). The SoNar sensor/cpYFP was excited at 420 and 485 nm and the emission at 520 nm was detected to obtain the ratiometric measurement. NAD$^+$/NADH ratios were obtained from the emission value of the 420 nm excitation (F420) and 485 nm excitation (F485). After background subtraction and normalization for cell number, obtained from absorbance at 600 nm in the same plate reader, the ratios of NAD$^+$/NADH were calculated as F420/F480. All samples were analyzed in triplicate.

## Acknowledgements

This research was funded by the NIH grant (RO1 DK41737) to SS, who holds a Tata Chancellor's Endowed Professorship in Molecular Biology. We thank Dr. Barry Bochner and In Iok Kong from Biolog Inc, Hayward, CA for advice and use of their Biolog machine. We also thank Agilent for the Seahorse demo unit, Dr. Anthony Molina for access to the Agilent Seahorse instruments at UCSD, and Dr. Stephan Dozier for advice using the Seahorse analyzer. Finally, we thank Dr. Y Yang and Dr. Cai (ECUST, Shanghai, China) for sequences of SoNar plasmids and *P. pastoris* strains deleted for the transcriptional repressors, respectively, and Dr. M Zhou (Shanghai, China) for *P. pastoris* kinase deletion strains.

## Additional information

### Funding

| Funder | Grant reference number | Author |
|---|---|---|
| National Institute of Diabetes and Digestive and Kidney Diseases | DK41737 | Suresh Subramani |

The funders had no role in study design, data collection, and interpretation, or the decision to submit the work for publication.

### Author contributions

Jean-Claude Farre, Conceptualization, Formal analysis, Investigation, Methodology, Validation, Visualization, Writing – original draft, Writing – review and editing; Krypton Carolino, Lou Devanneaux, Investigation; Suresh Subramani, Conceptualization, Formal analysis, Funding acquisition, Investigation, Supervision, Validation, Writing – review and editing

### Author ORCIDs

Jean-Claude Farre ⓘ http://orcid.org/0000-0001-8785-6474
Suresh Subramani ⓘ http://orcid.org/0000-0003-0180-1742

### Decision letter and Author response

Decision letter https://doi.org/10.7554/eLife.75143.sa1
Author response https://doi.org/10.7554/eLife.75143.sa2

## Additional files

### Supplementary files

- MDAR checklist

- Supplementary file 1. Strains and plasmids table.

- Supplementary file 2. RT-qPCR primers.

- Source data 1. Western blots and Ponceau S stained membranes (raw and annotated data).

- Source data 2. Western blots and Ponceau S stained membranes (raw and annotated data).

- Source data 3. Excel files containing the data for *Figure 2—figure supplement 1*, *Figure 4E*, *Figure 5*, and *Figure 5—figure supplement 1*.

### Data availability

All data generated or analyzed during this study are included in the manuscript and supporting file; source data files have been provided.

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
