## [Editor Report]

This paper elegantly describes how the assembly, division and proliferation of peroxisomes are controlled by proteins/metabolites residing in 4 different subcellular compartments. Mitochondrial energy production is assisted by peroxisomal proteins and metabolites, which shuttle via the cytosol to mitochondria, where ATP is produced. ATP availability is transmitted by the nuclear shuttling of cytosolic proteins to regulate genes controlling peroxisome assembly, division and proliferation.

---

## [Decision Letter]

[Editors' note: this paper was reviewed by Review Commons.]

---

## [Author Response]

Reviewer #1 (Evidence, reproducibility and clarity (Required)):Peroxisomes are non-autonomous organelles that receive their proteins from the cytosol and their lipids from the endoplasmic reticulum. In addition, they exchange their metabolites with several cellular compartments, in particular with mitochondria and the cytosol. Owing to this highly interdependent nature of peroxisomes, the regulation of peroxisomal proliferation is carefully controlled and balanced in eukaryotic cells. In the present study, the authors claim that they discovered "THE mechanism by which peroxisomal and mitochondrial metabolites influence redox and energy metabolism, while also influencing peroxisome biogenesis and proliferation, thereby exemplifying interorganellar communication and interplay involving peroxisomes, mitochondria, cytosol and the nucleus" (quoted from the abstract). However, in essence, they show data that yeast mutants in OXPHOS genes induce the Snf1 signaling (which is well known) and that Snf1-dependent signaling reactions are important for peroxisome function (which is also not novel). This study does not considerably increase our understanding of how peroxisomal and mitochondrial functions are coordinated. Identification of the individual contribution of specific proteins, lipids and metabolites that connect peroxisomal function to other processes is difficult. The authors make many assumptions about potential implications of their mutants, for example on the levels of NADH/NAD+, NADPH/NADP+ or ATP without measuring any of these. As it stands, this study is mainly descriptive and does not present the breakthrough the authors proposed. It therefore does not lead to the conceptual advance promised. It might however be suited for a specialized journal.

The reviewer has fundamentally misunderstood several key points, which makes us dispute the resulting judgment. Based on the reviewer’s statement that it is well known that yeast mutants in OXPHOS gene induce SNF1 signaling, and the fact that SNF1 signaling in oleate induces peroxisome biogenesis, one might have expected the induction of peroxisome biogenesis and proliferation in the OXPHOS mutants. Instead, we show the opposite effect – genes regulated by SNF1 signaling are downregulated in the OXPHOS mutants, despite our finding of Snf1 activation by phosphorylation due to the low ATP status of the cells. We found that Gal83 translocation to the nucleus and the predicted induction of peroxisome biogenesis, division and proliferation caused by Snf1 phosphorylation does not occur in the OXPHOS mutants. Surely, this is a fundamental misunderstanding on the part of the reviewer.

The reviewer claims incorrectly that Snf1-dependent signaling reactions are known, but this misses the point that despite Snf1 phosphorylation, we see severe defects in peroxisome biogenesis, division and proliferation in OXPHOS mutants. Moreover, the opposite mechanism that the reviewer claims is well known, wherein yeast OXPHOS mutants induce Snf1 signaling, is also not true. Only one publication describes this mechanism and was published this year (Liu et al., 2021). Most importantly, this study has fundamental differences from our paper – their assays were performed in glucose medium, a condition wherein peroxisome metabolism is not required and Snf1 is inactive. In their work, they showed that during glucose incubation, OXPHOS mutants activate Snf1 to upregulate glycolysis.

We disagree strongly with the statement that “this study does not considerably increase our understanding of how peroxisomal and mitochondrial functions are coordinated”. We show that NADH production in peroxisomes and the cytosol is necessary for OXPHOS, that OXPHOS and ATP production are necessary for peroxisome biogenesis, division and proliferation, and that this mechanism is mediated via Snf1, which despite being canonically activated by phosphorylation, does not result in nuclear translocation of the SNF1-complex subunit, Gal83. Finally, the peroxisome defects in the Δ*nugM* and Δ*gal83* mutants can be rescued by the deletion of three repressor genes that affect these processes (Figure 7). These findings do indeed “considerably increase our understanding of how peroxisomal and mitochondrial functions are coordinated” despite the reviewer’s contrarian view.

We agree with the reviewer that the measurement of NAD^+^/NADH ratios and ATP would strengthen the manuscript and have added these data (Figure 4B, C).

Major comments:1. The authors propose that the cooperation of peroxisomes and mitochondrial respiration is essential to maintain the cellular (cytosolic) NADH/NAD+ ratio. However, this ratio was not measured and assumptions are speculations on the basis of mutants lacking individual enzymes. Genetic probes to measure NADH redox potential (and levels) were recently introduced by several groups and optimized for studies in yeast. These reporters would make it possible to measure NADH redox levels precisely and in specific compartments. Data with such probes would make this study much more interesting for a general audience.

We agree that measurement of the NAD^+^/NADH ratios will be useful, but we do note that it is common in biology to use mutants proven to affect the production of known metabolites, so it is unfair to say that our conclusions are based on speculations, rather than knowledge of the biochemical pathways involved. Currently available genetic sensors for NAD^+^/NADH have not been applied yet to yeast*^1^*, or to peroxisomes. However, we successfully implemented SoNar probes in *P. pastoris* and we measured the NAD^+^/NADH ratios in the cytosol and peroxisomes in selected strains representing a good spectrum of this study, and these results are included (new Figure 5A). Since our model is that NAD^+^/NADH ratios affect ATP production, we adapted the Agilent Seahorse instrument, which measures oxygen consumption as a measure of mitochondrial ATP production, to show for the first time in *P. pastoris* that oxygen consumption rates (OCR) are indeed much lower in OXPHOS, AMPK, NADH-shuttling and *pex* mutants, relative to wild-type, and we show these data in Figure 5B.

2. The authors draw conclusions mainly from microscopic imaging of peroxisomes and from Western Blots with individual peroxisomal proteins. In order to be able to draw conclusions about complex gene expression patterns, for example to delineate the role of Snf1 signaling for peroxisome biogenesis, more unbiased omics datasets would be required. If the authors are not able to perform gene expression analyses they might simply use data previously published by others and focus specifically on genes relevant in the context of peroxisome proliferation. For example, the relevance of Snf1 signaling for mitochondrial function was analyzed in depth in a recent study published in EMBO reports (ref. 103 of this manuscript). This study contains RNAseq data-sets for Snf1-dependent gene expression (EV8) and the authors might just use these data.

We have indeed explored the possibility of additional insights through published ‘omics datasets, but concluded that they fail to do so at this time. The Snf1 dataset is available only for glucose-grown *S. cerevisiae^2^*, and is not so useful because we are looking at induction of peroxisome biogenesis, division and proliferation in *P. pastoris* grown in oleate or methanol. Just as an example, we know, as shown in Figure 4A, C and E, that *PEX11* and *POT1* are induced upon shift of *P. pastoris* from glucose to oleate, concurrently with Snf1 activation (Figure 4D), yet a comparison of the RNA-Seq analysis data from dataset EV8 in the paper from Liu et al., 2021.

Since the OXPHOS mutants are rescued by deletion of *MIG1*, *MIG2* and *NRG1*, we presume that our genes of interest are targets of this repression. Previous RNA-Seq analysis in *P. pastoris* has shown that the repressors Mig1, Mig2 and Nrg1 are involved in the glucose-mediated repression of genes involved in the metabolism of carbon sources other than glucose, during gluconeogenesis, Krebs cycle and respiration*^3, 4^*. It has also been shown in *P. pastoris*, that these repressors are involved in the downregulation of methanol utilization, peroxisome biogenesis and peroxisomal metabolic pathways, as well as the transcriptional activator, Mit1*^3, 4^*. We show here that, in addition, these repressors also repress peroxisome proliferation by an unknown mechanism that might be mediated by their direct or indirect targets. Indeed, about 23% (1197 genes) of *P. pastoris* are downregulated in the Δ*mig1* Δ*mig2* strain and 7% (357 genes) are upregulated*^3^*. Our finding that the triple mutant (Δ*mig1* Δ*mig2 Δnrg1*) strain rescues the peroxisome biogenesis and proliferation defects of the Δ*nugM* mutant could be accounted for by the induction of one or more of these genes, but the list of genes upregulated by the absence of the repressors is too long, making it challenging to validate whether any of these is responsible for peroxisome proliferation in a reasonable time frame.

3. The authors make many strong statements about the relevance of their observations based on very indirect assumptions. The authors should downgrade these statements. The abstract should summarize the direct conclusions from experiments. Bold sentences should be removed such as "We uncovered here the mechanism by which peroxisomal and mitochondrial metabolites influence redox and energy metabolism, while also influencing peroxisome biogenesis and proliferation, thereby exemplifying interorganellar communication and interplay involving peroxisomes, mitochondria, cytosol and the nucleus." (Abstract) This is simply not true! The take-home message is rather 'gene expression is extremely complex, can be regulated at many levels and often includes the integration of many signals' (quoted from page 20). However, unfortunately, this study makes the picture not much clearer than it was before.

We disagree because the reviewer has based these judgements on an incorrect understanding of our paper. The abstract describes our experimental observations, but we have replaced “we uncovered here” with “Our results are interpreted in terms” to accommodate the reviewer.

Reviewer #1 (Significance (Required)):Nature and significance of the advance:Collection of descriptive data. Advance is limited.

We disagree respectfully, as described above, but appreciate the suggestions for improvement of the manuscript.

Confirms previous conclusions on the relevance of Snf1 signaling.

We disagree for reasons stated above.

Reviewer #2 (Evidence, reproducibility and clarity (Required)):Summary: This study elucidates the acting points of mitochondrial OXPHOS activity on peroxisome proliferation. The metabolite of peroxisome activities, namely NADH, was found to be intervened in the peroxisome-mitochondrion relay of cellular metabolic status. ATP seemed to be the output of mitochondrial activity to support the peroxisome proliferation, since the uncoupler exerted the same inhibitory effects as loss of the NADH shuttling machinery. Blockings of these pathways affected the intra-cellular dynamics of Snf1-kinase complex as evident by Gal83 localization experiments. The inhibitory effects on the peroxisome proliferation by the OXPHOS deficiency were canceled by further deletions of MIG1, MIG2, and NRG1 genes, all of which are transcriptional repressors regulated by Snf1 complex. By contrast, an artificial boosting of Mxr1-Mit1 pathway did not suppress the inhibitory effects.Major comments:The huge story of this study is well supported by the elaborately arranged experimental data and this study can be highly evaluated as elucidation of the SNF1 function in glucose derepression mechanism. Since most experiments were performed under artificial conditions, it is unclear whether the peroxisomes-mitochondria interplay functions under physiological conditions. What is the physiological significance of downregulating peroxisome proliferation by dysfunction of mitochondria? Indeed, deletion of three transcriptional repressors rescued the phenotype of mitochondrial dysfunction, but it seems that there is no physiological significance because WT cells have these repressors and these just bind promoter regions and repress gene expression. The authors used the term "feedback loop" in this manuscript, but the feedback loop cannot occur under physiological conditions.

We appreciate the first sentence, which is also supported by Reviewer 3, but contradicts the view of Reviewer 1.

We agree that the physiological significance of the inappropriate and appropriate induction of peroxisome biogenesis, division and proliferation is important and may not have been stated clearly in our manuscript. The induction and proliferation of peroxisomes requires ATP, as well as the availability of substrates for peroxisomal metabolic pathways. In their absence, one would expect no induction of peroxisome functions. The physiological consequence of deleting *MIG1*, *MIG2* and *NRG1* (i.e. inappropriate peroxisome induction) is published, but had not been cited by us, which is now corrected. Both the Δ*nrg1* and the Δ*mig1* Δ*mig2* mutants do have a fitness cost, as expected, during growth in glucose and glycerol, and less so in methanol*^3, 4^*.

The physiological consequences of the inability to appropriately induce peroxisome functions when needed (in oleate and methanol) has been described in the paper along with the pathway involved, which is new. OXPHOS mutants are unable to grow in oleate and methanol, and also cannot divide or proliferate peroxisomes. We clearly disagree with the reviewer that this physiological significance is unclear.

To be clear, we are not studyng how cells are adapting in the OXPHOS mutants but rather, we are trying to study a mechanism occuring in the WT cells, where mitochondrial ATP is produced due to peroxisome metabolism and this induces peroxisome proliferation. The data presented for Δ*pex14* cells, in which OXPHOS is fine, but peroxisome proliferation is induced by lactate, indicates that we are studying a physiological mechanism.

The reviewer may not fully understand what we mean by the term “feedback loop”. We believe that in oleate and methanol, NADH production by peroxisomes is necessary for mitochondria to produce ATP, which in turn is required for Snf1/Gal83 activation and induction of peroxisome biogenesis, division and proliferation. These are all happening under physiological conditions of oleate and methanol growth. We hope that our explanation provides a clearer picture of why we disagree with the reviewer statement that “feedback loop cannot occur under physiological conditions.” The feedback loop is explained explicitly in the revision.

Another concern is that the growth curves of various mutant strains were not presented. Mitochondrial dysfunction should have affected the growth on non-fermentative carbon sources and ATP depletion should have affected various basic cellular processes. To know the level of growth defect by each strain, the growth curves of all mutant strains on glucose, oleate, and methanol should be presented in the supplementary.

While this is easily done, it is busy work that does not add value and it is irrelevant to request this unnecessary information for all strains. Any mutant strains without functional mitochondria will not grow on non-fermentable carbon sources. The reviewer asks for growth information in methanol and oleate, but perhaps misses the point that in the absence of key metabolic enzymes like alcohol oxidase (Aox1) and thiolase required for their utilization, and in the absence of ATP production, there is no growth of the OXPHOS mutants in these media. This information has been added.

Reviewer #2 (Significance (Required)):"Feedback loop" has not been verified by experiments, and the manuscript does not contain new new concepts, and the obtained results are within the expected range.

The reviewer may not have understood what we meant by “feedback loop”, which is explained above and in the text. We believe we have indeed verified the steps of this loop – i.e. the requirement of NADH produced by mitochondria, peroxisomes and NADH shuttles, mitochondrial ATP production, SNF1 activation including Gal83 nuclear translocation and removal of repression by MIG1, MIG2 and NRG1 for peroxisome biogenesis, division and proliferation.

Referee Cross-commentingI mostly agree with Reviewer #1's opinion.The manuscript contains too many speculation, which should be verified experimentally. As I mentioned in the review, ATP level as well as redox metabolites should be determined. And "Feedback loop" should be verified step by step experimentally.

See our response above. New ATP measurements are in Figure 5B.

Reviewer #3 (Evidence, reproducibility and clarity (Required)):The manuscript “OXPHOS deficiencies affect peroxisome proliferation by downregulating genes controlled by the SNF1 signaling pathway" submitted by Farré et al. describes a metabolic signaling network between peroxisomes and mitochondria adjusting peroxisomal abundance to the energetic status of the cell. The authors show, that NADH shuttling from peroxisomes to mitochondria is required to induce peroxisome proliferation when Pichia pastoris is grown on nutrients which are metabolized in peroxisomes (methanol, oleate). Thus, regulation of peroxisome abundance is not directly linked to peroxisomal NADH release but to mitochondrial ATP production by OXPHOS as shown by complex I and complex III mutant strains and uncoupling of ATP production from electron transport chain. As SNF1 is a kinase complex regulating cellular gene expression in response to the cellular ATP status, the authors further analyzed the SNF1-dependency of the observed phenomena. Interestingly, the authors show that the effect of mitochondrial OXPHOS disruption does not take place at the level of SNF1-activation by phosphorylation, but further downstream in the signaling pathway: activated SNF1 is prevented from shuttling to the nucleus in OXPHOS mutant strains, where it would normally induce the release of transcriptional repressors from the promotors of genes for peroxisome proteins. Unfortunately, one last issue – how Gal83-mediated SNF1 nuclear shuttling is altered by the cellular ATP status remains unresolved, which somewhat reduces the impact of the study.In general, the manuscript is soundly written, most largely convincing and of a fair experimental quality with adequately replicated experiments and adequate statistical analysis. Before manuscript acceptance, I would suggest, however, some additional control experiments to strengthen further the conclusions drawn by the authors.

We appreciate the comments and feel that this reviewer does indeed understand fully the significance of our study. We agree that understanding “how Gal83-mediated SNF1 nuclear shuttling is altered by the cellular ATP status” would add to the impact but this is beyond the scope of a few quick experiments.

Major comments:1. While the authors effectively block NADH shuttling from peroxisomes to mitochondria, transport of acetyl-CoA produced by peroxisomal β-oxidation of oleate should still be possible via the carnitine shuttle system. In this regard, it is surprising that the ΔmdhB strain and ΔgpdA/ΔmdhB strain show such a considerably strong phenotype as one round of the TCA cycle with peroxisome-derived acetyl-CoA could produce considerably more intramitochondrial NADH as generated by a single round of chain-shortening via peroxisomal β-oxidation (theoretically 3:1). A peroxisomal CRAT (carnitine acetyltransferase) deletion strain could prevent that acetyl-CoA is shuttled out of peroxisomes to be metabolized by the mitochondrial TCA cycle and would be an important additional control. Alternatively, measurement of cytosolic and/or mitochondrial ATP levels could give information how effectively the block of NADH shuttling from peroxisomes compromises mitochondrial ATP production.

We investigated the CAT2 (peroxisomal CRAT) deletion and found, as hypothesized by the reviewer, that the *cat2Δ* strongly inhibited peroxisome proliferation in oleate and surprisingly also in methanol (new Figure 1 E,F). A recent *P. pastoris* study showed that Cat2 is strongly induced when methanol/glycerol mix is used as the only carbon source and might have a significant role in the methanol assimilatory pathway, which feeds into the TCA cycle*^6^*. We did not observe a major difference in ATP productions (OCR values in new Figure 5B) between deletions impairing NADH-shuttling proteins and Cat2, but both were significantly lower than wild-type, suggesting that both pathways contribute to ATP production.

2. Figure 1D: BFP-SKL signals in the ΔgpdA/ΔmdhA strain and ΔgpdA/ΔmdhB strains show a relatively high cytosolic background signal. Does this cytosolic fluorescence signal indicate an incomplete import of matrix proteins in these strains, when cultured on methanol, or is this just unspecific staining?

It is due to incomplete import due to the use of the strong and constitutive *GAP* promoter to drive BFP-SKL.

3. Figure 2B, C: While the complex I and III deficient mutants show a good colocalization between BFP-SKL and Pex3-GFP signals, when grown on methanol, all three mutants show bright GFP-SKL signals but appear to be devoid of a specific Pex3-GFP signal, when cultivated on oleate, which is however, not mentioned in the corresponding manuscript text (page 15). In the immunoblots in Figure 4C+D, endogenous Pex3 expression in the complex I ΔnugM strain appears to be unchanged when grown on oleate. The authors should comment on these discrepant findings, since altered Pex3-GFP expression would likely influence peroxisome proliferation capacities in the mutant strains. To clarify this issue, further experimental data on Pex3-GFP levels in the complex I/III mutant strains grown on oleate might be required. Furthermore, it is quite surprising, that according to Figure 4D, Pex3 expression levels do not change in response to methanol- or oelate-induced peroxisome proliferation. This would either mean, that the peroxisomes in glucose grown Pichia (low PO abundance) would exhibit unexpectedly high membrane surface densities of Pex3 if compared to peroxisome under proliferation or that a considerable amount of Pex3 might be localized at the ER, to facilitate a rapid proliferation in response to nutrient change. The authors should add an appropriate comment to the Discussion section.

In *S. cerevisiae,* PMPs are not induced as much as matrix proteins*^7^* and similarly in *P. pastoris*, Aox1 can represent more than 50% of cell’s protein. Pex3 is present in glucose media in reasonable amounts and is induced only few-fold several hours after the shift to methanol or oleate in WT or the Δ*nugM* mutant (Figure 4D). However, proteins such as thiolase, Aox1 and Pex11 are fully repressed in glucose and are then strongly induced in oleate and methanol (new Figure 4, figure supplement 1A). Experiments performed in Figure 4 were after 4 hours in the respective media. It is worth noting that the short induction time was chosen to observe Snf1 phosphorylation, which disappears after longer induction in methanol or oleate. Pex3-GFP localization in OXPHOS mutants can be easily observed after longer inductions in oleate (16 h instead of 8h). Figure 2A was replaced with data after a longer induction in oleate.

4. The authors use an antibody against mammalian phospho-AMPKα to analyze the phosphorylation of SBF1 by immunoblotting. Even if the protein sequence in the correspondent ppSNF1 region is well preserved for such a considerable evolutionary distance, the antibody's potential to specifically recognize only phosphorylated SNF1 should be experimentally verified, since further less conserved regions of SNF1 might influence the protein's general structure. Using a phosphatase treatment in lysates before or after blotting could act as an appropriate control to be presented in a supplemental figure.

We note that the antibody has extensively been used in *S. cerevisiae*, where they confirm the phosphorylation status*^8, 9^* and PpSnf1 is an identical protein in this activation loop. This has been added to the text. But we confirmed the specificity by dephosphorylation of proteins fixed on a membrane and included the data as new Figure 4, figure supplement 1B.

5. A key finding to explain the failure of mig1/mig2/nrg1 de-repression by activated (phospho-)SNF1 in OXPHOS mutants is the inhibition of a Gal83-mediated nuclear SNF1 shuttle in response to ATP-depletion. Unfortunately, the figures of the nuclear localization of Gal83-GFP grown on oleate are hardly convincing. The fluorescent foci in Figure 7 show GFP-signals that appear adjacent to the nucleus but lack the resolution to proof a localization at specific foci within the nucleus.

We think this in unnecessary because we make no claim about any subnuclear localization of Gal83.

Likewise, all cells with a strong Sec61-RFP/Gal83-GFP colocalisation in Figure S5 appear to be according to the DIC image not in a central focal plane. High resolution images combined with fluorescent nuclear and cytosolic markers should be provided to proof that Gal83-GFP is effectively shuttled into the nucleus in WT Pichia grown on oleate. Alternatively, the authors might analyze that significant fractions of Gal83-GFP enter the nucleus by a subcellular fractionation approach, separating nuclei from cytosol and a microsomal fraction (signals for Gal83-GFP might indicate e.g. a localization to perinuclear ER marked by the Sec61-mCherry probe).

We agree and replaced the previous Gal83-GFP localization ((old Figure 7) by a new localization (new Figure 8) but in the presence of Leptomycin B (LMB), a potent and specific Crm1 nuclear export inhibitor, required for Gal83 nuclear export in *S. cerevisiae^10^*). The presence of the inhibitor allowed us to provide better evidence of Gal83-GFP shuttling to the nucleus in WT cells during peroxisome proliferation conditions and the lack of this shuttling in the mitochondrial mutant.

Reviewer #3 (Significance (Required)):Currently, there is increasing evidence that organelles have to communicate directly via physical contacts or indirectly across the cytosol with each other to maintain cellular homeostasis. Peroxisomes, which are highly dynamic organelles specialized in lipid catabolism and in yeast species methanol oxidation, have to be adjusted in number and protein composition to the current metabolic state of the cell. Despite this obvious need for organelle plasticity, there is still only scarce information on the cellular signaling networks, which control peroxisome abundance within a cell. Moreover, peroxisomes do seldom house complete metabolic pathways but have to exchange metabolic intermediates with other cellular compartments such as the endoplasmic reticulum or mitochondria. Of note, mitochondria are often found to be compromised in patient cells with peroxisomal gene defects underlining the strong functional relationship between both organelles. In addition to their metabolic cooperation in e.g. β-oxidation of fatty acids or catabolism of reactive oxygen species, both organelles are linked at the level of their fission pathways, as they use an overlapping protein assembly for this process. Therefore, the manuscripts presents an intriguing finding and extension to the concept of an intricate communication between peroxisomes and mitochondria, that peroxisomal abundance can be controlled via a signaling network regulated by mitochondrial ATP production in the yeast Pichia pastoris. In this context, the manuscript nicely exemplifies how a relatively simple but straightforward feedback mechanism is able to effectively adjust pathways of energy production to current nutrient availability via an SNF1-dependent ATP-sensitive cross-organellar signaling network.

This reviewer clearly spells out the key points of our paper accurately and fairly. We appreciate the careful analysis.

Since mammalian cells possess with AMPKA a direct orthologue of SNF1, the authors speculate, that a similar signaling network linking peroxisome abundance to mitochondrial energy production could likewise exist in humans. However, it should be taken into consideration that unlike in yeast species, which exclusively metabolize fatty acids and methanol in peroxisomes for energy production, their counterparts in mammalian cells are of low abundance. Thus, they are able to contribute only with a minor percentage to net cellular energy production but are rather organelles specialized on metabolizing low abundant very long chain and branched chain fatty acids, which are in higher concentrations toxic for cells. Therefore, the direct transferability of the findings to mammals might be limited and need further evaluation. Nevertheless, even if a direct extrapolation of the findings to the human situation might not be adequate, the manuscript still introduces a principal and exemplary mechanism regulating cellular homeostasis via an organellar communication network and is thus of general interest for the cell biological community.

We agree.

In summary, we have dealt with all of the questions and suggestions of the reviewers. We hope you find the manuscript ready for acceptance.

References

[1] Zhao, Y., Hu, Q., Cheng, F., Su, N., Wang, A., Zou, Y., Hu, H., Chen, X., Zhou, H. M., Huang, X., Yang, K., Zhu, Q., Wang, X., Yi, J., Zhu, L., Qian, X., Chen, L., Tang, Y., Loscalzo, J., and Yang, Y. (2015) SoNar, a highly responsive NAD^+^/NADH sensor, allows high-throughput metabolic screening of anti-tumor agents, *Cell Metab 21*, 777-789.

[2] Liu, S., Liu, S., He, B., Li, L., Li, L., Wang, J., Cai, T., Chen, S., and Jiang, H. (2021) OXPHOS deficiency activates global adaptation pathways to maintain mitochondrial membrane potential, *EMBO Rep 22*, e51606.

[3] Shi, L., Wang, X., Wang, J., Zhang, P., Qi, F., Cai, M., Zhang, Y., and Zhou, X. (2018) Transcriptome analysis of Δ*mig1* Δ*mig2* mutant reveals their roles in methanol catabolism, peroxisome biogenesis and autophagy in methylotrophic yeast *Pichia pastoris*, *Genes Genomics 40*, 399-412.

[4] Wang, X., Cai, M., Shi, L., Wang, Q., Zhu, J., Wang, J., Zhou, M., Zhou, X., and Zhang, Y. (2016) PpNrg1 is a transcriptional repressor for glucose and glycerol repression of *AOX1* promoter in methylotrophic yeast Pichia pastoris, *Biotechnol Lett 38*, 291-298.

[5] Nguyen, L. N., Gacser, A., and Nosanchuk, J. D. (2011) Secreted lipases supply fatty acids for yeast growth in the absence of de novo fatty acid synthesis, *Virulence 2*, 538-541.

[6] Camara, E., Landes, N., Albiol, J., Gasser, B., Mattanovich, D., and Ferrer, P. (2017) Increased dosage of *AOX1* promoter-regulated expression cassettes leads to transcription attenuation of the methanol metabolism in Pichia pastoris, *Sci Rep 7*, 44302.

[7] Wiemer, E. A., Luers, G. H., Faber, K. N., Wenzel, T., Veenhuis, M., and Subramani, S. (1996) Isolation and characterization of Pas2p, a peroxisomal membrane protein essential for peroxisome biogenesis in the methylotrophic yeast *Pichia pastoris*, *J Biol Chem 271*, 18973-18980.

[8] McCartney, R. R., Garnar-Wortzel, L., Chandrashekarappa, D. G., and Schmidt, M. C. (2016) Activation and inhibition of Snf1 kinase activity by phosphorylation within the activation loop, *Biochim Biophys Acta 1864*, 1518-1528.

[9] McCartney, R. R., and Schmidt, M. C. (2001) Regulation of Snf1 kinase. Activation requires phosphorylation of threonine 210 by an upstream kinase as well as a distinct step mediated by the Snf4 subunit, *J Biol Chem 276*, 36460-36466.

[10] Hedbacker, K., and Carlson, M. (2006) Regulation of the nucleocytoplasmic distribution of Snf1-Gal83 protein kinase, *Eukaryot Cell 5*, 1950-1956.